# Ozone Impacts of Gas-Aerosol Uptake in Global Chemistry Transport Models

Scarlet Stadtler[1], David Simpson[2,3], Sabine Schröder[1], Domenico Taraborrelli[1], Andreas Bott[4], and Martin Schultz[1]

[1]Institut für Energie- und Klimaforschung, IEK-8, Forschungszentrum Jülich, Germany
[2]EMEP MSC-W, Norwegian Meteorological Institute, Oslo, Norway
[3]Dept. Space, Earth & Environment, Chalmers University of Technology, Gothenburg, Sweden
[4]Meteorological Institute University Bonn, Bonn, Germany

*Correspondence to:* David Simpson (david.simpson@met.no)

**Abstract.**

The impact of six heterogeneous gas-aerosol uptake reactions on tropospheric ozone and nitrogen species was studied using two chemical transport models, EMEP MSC-W and ECHAM-HAMMOZ. Species undergoing heterogeneous reactions in both models include $N_2O_5$, $NO_3$, $NO_2$, $O_3$, $HNO_3$ and $HO_2$. Since heterogeneous reactions take place at the aerosol surface area, the modeled surface area density $S_a$ of both models was compared to a satellite product retrieving the surface area. This comparison shows a good agreement in global pattern and especially the capability of both models to capture the extreme aerosol loadings in East Asia.

The impact of the heterogeneous reactions was evaluated by the simulation of a reference run containing all heterogeneous reactions and several sensitivity runs. One reaction was turned off in each sensitivity run to compare it with the reference run. The analysis of the sensitivity runs confirms that the globally most important heterogeneous reaction is the one of $N_2O_5$. Nevertheless, $NO_2$, $HNO_3$ and $HO_2$ heterogeneous reaction gain relevance particularly in East Asia due to the presence of high $NO_x$ concentrations and high $S_a$ in the same region. ~~although ECHAM-HAMMOZ showed much stronger responses than EMEP in this respect.~~ The heterogeneous reaction of $O_3$ itself on dust is of minor relevance compared to the other heterogeneous reactions. The impacts of the $N_2O_5$ reactions show strong seasonal variations, with biggest impacts on $O_3$ in spring time when photochemical reactions are active and $N_2O_5$ levels still high. Evaluation of the models with northern hemispheric ozone surface observations yields a better agreement of the models with observations in terms of concentration levels, variability, and temporal correlations at most sites when the heterogeneous reactions are incorporated. Our results are loosely consistent with results from earlier studies, although the magnitude of changes induced by $N_2O_5$ reaction is at the low end of estimates, which seems to fit a trend whereby the more recent the study the lower the impacts of these reactions.

## 1 Introduction

Nitrogen species, ozone and atmospheric aerosols are major pollutants in the atmosphere, having strong impacts on ecosystems and human health, and also interacting with climate (Ainsworth et al., 2012; Harrison and Yin, 2000; Simpson et al., 2014;

IPCC, 2013). In regions, where gas phase and aerosol pollutants meet, heterogeneous chemistry can play a significant role (Jacob, 2000). The first heterogeneous process to become prominent in atmospheric chemistry was the heterogeneous destruction of stratospheric ozone on polar stratospheric clouds (Solomon et al., 1986; Solomon, 1999). However, heterogeneous processes are also relevant in the lower atmosphere, influencing tropospheric ozone and therefore oxidation capacity of the atmosphere

(Ravishankara, 1997; Pöschl, 2005; Seinfeld and Pandis, 2012). An important example is the heterogeneous reaction of $N_2O_5$ on aerosols, which is known to impact the $NO_x$-$O_3$ cycle while mainly removing $NO_x$ from the troposphere (Mozurkewich and Calvert, 1988; Dentener and Crutzen, 1993; Evans and Jacob, 2005; Chang et al., 2011; Brown and Stutz, 2012), which can lead to ozone reduction (Macintyre and Evans, 2010). Other oxidised nitrogen species also undergo heterogeneous reactions on different aerosol types. $NO_2$, $HNO_3$ and $NO_3$ react on wet surfaces of different aerosol types and increase aerosol nitrate

content (Rudich et al., 1998; Goodman et al., 1999). $HNO_3$ reacts also with dust and sea salt particles which is again a sink for $NO_x$ and a source for particulate nitrate (Davies and Cox, 1998; Hanisch and Crowley, 2001). Moreover, heterogeneous reaction of $NO_2$ produces HONO which plays the role of a reservoir specie for NO and OH production (Platt et al., 1980). Other species also undergo heterogeneous reactions. $O_3$ reacts on dust particles and this has been estimated to lead to an ozone loss of about 20% in dusty regions (Usher et al., 2003). $HO_2$ reacts on wet particles leading to $H_2O_2$ production (Thornton

and Abbatt, 2005). Furthermore, heterogeneous reactions lead to halogen release from sea salt aerosols (Frenzel et al., 1998; Yang et al., 2008; Lowe et al., 2011). Many modeling studies have been conducted over the years on these processes, but usually heterogeneous reactions were studied individually, and typically considering annual global budgets rather than detailed temporal or spatial resolution of the impacts (Dentener and Crutzen, 1993; Rudich et al., 1998; Saathoff et al., 2001; Bauer et al., 2004; Hodzic et al., 2006; Thornton et al., 2008; Chang et al., 2011).

This paper presents estimates of the global impact of heterogeneous reactions of $N_2O_5$, $NO_3$, $NO_2$, $HNO_3$, $HO_2$ and $O_3$ and evaluates each reaction in a systematic way. The influence of each reaction on the magnitude and spatial and temporal variation in surface ozone is illustrated. The greatest impacts are seen in northern hemispheric regions of North America, Europe, South and East Asia. The $N_2O_5$ reaction is shown to significantly affect the spring-peak of surface $O_3$ at sites in all these regions. ~~Although the impact of $N_2O_5$ reaction on $O_3$ is analysed, due to technical limitations in both models no $ClNO_2$ chemistry is~~

~~included, which could decrease the impact of $N_2O_5$ on $O_3$, since it is a competing $NO_x$ loss process. Different limitations in both model formulations are imposed here, as the lack of halogen chemistry and no possibility to transfer chlorine from the particle phase in the gas phase.~~

Sect. 2 presents the two global scale chemical transport models, EMEP MSC-W and ECHAM-HAMMOZ, as well as details of the reaction parameterisations and sensitivity tests. In Sect. 3 a short review of the range of reaction probabilities for each

heterogeneous reaction is given. Model setups and sensitivity runs are described in Sect. 4. Sect. 5 first presents a comparison of the simulated surface area from the models with satellite derived product, since the surface area of aerosols is crucial for heterogeneous chemistry. Especially in polluted regions where high trace gas concentrations meet large surface areas provided by aerosols heterogeneous chemistry might be of significant importance explaining aerosol composition and trace gas mixing ratios (Jacob, 2000; Pathak et al., 2009). Sect. 5 also presents the results of the sensitivity tests, and comparisons of daily

maximum ozone time-series for 2012 with surface station observations for selected sites. In Sect. 5.4 we discuss these results

compared to previous studies, as well as commenting on a number of open questions concerning heterogeneous reactions. Finally, Sect. 6 summarizes the results and implications for atmospheric chemistry.

## 2 Model description

Two models, the chemical transport model EMEP MSC-W (v4.16) (Simpson et al., 2012, 2017) and the global chemistry aerosol climate model ECHAM6.3-HAM2.3MOZ1.0 (Schultz et al., in preparation) were used to study the heterogeneous chemistry of various compounds in the atmosphere.

### 2.1 EMEP

The basis of the EMEP MSC-W chemical transport model has been described in detail by Simpson et al. (2012), but substantial updates have been made in the treatment of aerosols, biogenic emissions and chemistry in recent years. Simpson et al. (2015, 2017) have documented the main changes in aerosol surface area and biogenic VOC emissions as discussed below, up to version rv4.15. The model version used in this report, rv4.16, is the latest version of the model. The main changes in rv4.16 have been the inclusion of the dry and wet deposition for $N_2O_5$, using the same rates as for $HNO_3$, and the implementation of an improved radiation scheme, based upon Weiss and Norman (1985). These changes have not affected basic model performance very much compared to previous publications, but of course the concentrations of $N_2O_5$ and its impact on ozone are reduced somewhat compared to earlier model versions.

The default model setup includes 20 vertical layers up to 100 hPa, using terrain-following coordinates, and the lowest layer has a thickness of about 90 m. Although originally designed for European applications (previously using a grid of resolution 50 km, more recently 28 km), the model is very flexible and is now applied on scales ranging from global (Jonson et al., 2010) to local (1-7 km grids), e.g. Vieno et al. (2010, 2014), Schaap et al. (2015). Anthropogenic emissions from land-based sources are here taken from the so-called PANHAM database from the EU PANDA project (http://panda-project.eu), which combined emissions from the global HTAP data base (http://edgar.jrc.ec.europa.eu/htap_v2/index.php?SECURE=123) with the MEIC database for China (http://www.meicmodel.org/).

Emissions of VOC from biogenic sources are calculated in the model based upon land-cover and meteorological conditions. Emission factors for earlier versions of the EMEP model were mainly intended for European simulations (Simpson et al., 1999, 2012), but during 2016-2017 the factors used in non-European areas were substantially revised - see Simpson et al. (2017) for details. For details of other emissions (soil-NO, lightning, aircraft, biomass-burning), see Simpson et al. (2012). For the present study meteorological data from the European Centre for Medium Range Weather Forecasting Integrated Forecasting System (ECMWF-IFS) model (http://www.ecmwf.int/research/ifsdocs/) were used, and the model runs with $1 \times 1°$ latitude-longitude resolution.

The chemical scheme in the EMEP MSC-W model, denoted 'EmChem16', consists of a standard gas-phase mechanism (132 species, 183 reactions, a recent update of the earlier EmChem03 evaluated by Andersson-Sköld and Simpson 1999), extended with organic aerosols using a volatility-basis-set scheme (Bergström et al., 2012; Simpson et al., 2012), plus sea-salt

(Tsyro et al., 2011) and dust aerosol. Unlike ECHAM-HAMMOZ, the EMEP model includes $NH_3$ and handles the resulting interactions with sulphate, $HNO_3$ and ammonium-nitrate through the use of the MARS equilibrium solver (Binkowski and Shankar, 1995). Unfortunately, interactions with sea-salt have not yet been implemented in EMEP-MARS. The chemical equations are solved using the TWOSTEP algorithm (Verwer and Simpson, 1995; Verwer et al., 1996).

The EMEP MSC-W model has been extensively compared with measurements of many different compounds with generally good performance (e.g. Simpson et al., 2006a, b; Fagerli and Aas, 2008; Aas et al., 2012; Gauss et al., 2011), although most of these studies have focused on Europe. Still, in comparisons with global data and other models, the EMEP MSC-W model seems to perform well, especially more recent versions (Jonson et al., 2010, 2015; Angelbratt et al., 2011; Bian et al., 2017).

As of EMEP MSC-W model version rv4.7 (Simpson et al., 2015), aerosol surface area ($S_a$) is estimated using the em-
pirical relations of Gerber (1985), which simply requires aerosol mass concentrations and assumed aerosol density and size-parameters. Values of $S_a$ are calculated for fine and coarse particulate matter ($PM_f$, $PM_c$) both as totals (including all components, for reaction R1-3,5 in Table 1), and separately for coarse sea-salt and dust particles - which we denote as $S_{ss}$ and $S_{du}$ respectively. The distinction between total area $S_a$ and $S_{ss}$ and $S_{du}$ was made to allow the use of Gerber's specific parameterizations for sea-salt and dust for reactions R5 and R6 (Table 1), with the assumption that where concentrations are large
(eg over oceans, deserts) these give a better estimate of $S_a$ than the rural parameterisation would give. Further, for $S_{du}$ the aerosol is assumed to be dry; which is not always true but is intended to reflect the nature of desert dust dominated aerosol. The EMEP model does not include fine-mode formation of $NO_3^-$ through reaction R4, since the relationship between $HNO_3$ and fine-mode nitrate is given by the thermodynamic equilibrium solver MARS.

## 2.2   ECHAM-HAMMOZ

ECHAM-HAMMOZ is an aerosol chemistry climate model capable of performing interactive aerosol chemistry simulations. For this study simulations were done using version ECHAM6.3-HAM2.3MOZ1.0 (https://redmine.hammoz.ethz.ch/projects/ hammoz/wiki/Echam630-ham23-moz10). The model system ECHAM-HAMMOZ consists of the general circulation model ECHAM6.3 (Stevens et al., 2013), the aerosol model HAM2.3 (Neubauer et al., in preparation; Zhang et al., 2012) and the chemistry model MOZ1.0 (Schultz et al., in preparation). ECHAM calculates meteorological variables, cloud processes and
radiative transfer considering greenhouse gases and aerosols. The simulations in this study use hybrid sigma coordinates with 47 vertical layers, while the surface layer thickness is about 50 m. The horizontal resolution T63 leads to an associated $1.875° \times$ 1.875 °Gaussian grid.

HAM simulates the evolution of aerosols considering aerosol and aerosol precursor emissions, microphysical processes as nucleation, coagulation, accumulation, sedimentation, dry and wet deposition. Via direct and indirect aerosol effects a feedback
to climate system is simulated (Neubauer et al., in preparation). The aerosols in HAM are assumed to be internally mixed and consist of up to 5 components: sulphate, sea salt, dust, organic carbon and black carbon. To describe the aerosol number the microphysical driver M7 uses distribution seven log normal functions describing four wet aerosol modes and three dry aerosol modes. Hence, the wet functions cover nucleation, Aitken, accumulation, and coarse modes and the dry functions do not cover the nucleation mode. The height and median radius of the distribution are calculated, just its width is fixed. Due to aerosol

aging it is possible for insoluble particles to become soluble (Vignati et al., 2004). Dust and sea salt emissions are interactively calculated considering the wind speed at 10 m. Dimethylsulphate emissions are parametrized and emissions of sulphate dioxide, sulphate aerosol, black carbon and organic carbon are taken from the Representative Concentration Pathway (RCP) 8.5 emissions (Van Vuuren et al., 2011). Finally, optical properties of the aerosol are calculated and impact the atmospheric
circulation in ECHAM (Zhang et al., 2012).

Atmospheric chemistry is simulated by MOZ which is based on MOZART3.5 (Model for Ozone and Related chemical Tracers version 3.5) (Stein et al., 2012) connecting tropospheric chemistry of MOZART4 (Emmons et al., 2010) and stratospheric chemistry of MOZART3 (Kinnison et al., 2007). Further development since Stein et al. (2012) lead to MOZ being a chemical mechanism resembling to CAM-chem (Community Atmosphere Model with Chemistry) (Lamarque et al., 2012) with sev-
eral revisions, extended chemistry of aromatic compounds and a more detailed isoprene chemistry based on Taraborrelli et al. (2009). The version MOZ1.0 used here consists of 242 tracers, 733 chemical reactions which contain 142 photolysis reactions, 6 heterogeneous tropospheric reactions and 16 stratospheric heterogeneous reactions. Further, MOZ calculates dry and wet deposition of gases. Anthropogenic emissions are taken from the emission inventory RCP 8.5 (Van Vuuren et al., 2011). Biogenic emissions of VOC and $NO_2$ are calculated interactively by MEGAN (Model of Emissions of Gases and Aerosols from Nature)
(Guenther et al., 2006; Henrot et al., 2017). NO lightning emission are parametrized as described by Grewe et al. (2001).

HAM and MOZ interact via two physical processes. First, assuming spherical aerosols, the surface area density for heterogeneous reactions is calculated using aerosol distribution and median radius. Second, MOZ provides fields of oxidants for aerosol formation from gas-phase precursors. The HAMMOZ coupling does not include ammonium nitrate formation due to the lack of nitrate aerosol in the current HAM version. Therefore, reactive uptake of nitric acid leads to a total loss, based on the assump-
tion of a quick loss of gas phase $HNO_3$ and particulate nitrate. To underline, heterogeneous reactions in ECHAM-HAMMOZ do not form $HNO_3$ in the gas phase, but introduce a direct loss to the products $HNO_3$ and $NO_3^-$.

## 3 Heterogeneous reactive uptake

Experimental studies show that oxidised nitrogen species, ozone and the hydroperoxy radical undergo heterogeneous reactions on wet and dry aerosols. Heterogeneous reactions can be modeled as a pseudo-first order process (Ammann et al., 2013).

$$\frac{d[X]_g}{dt} = -k_X[X]_g \tag{1}$$

The change in gas phase concentration of the species X = $N_2O_5$, $NO_3$, $NO_2$, $HNO_3$, $HO_2$, $O_3$ is proportional to its gas phase concentration $[X]_g$ and a reaction rate coefficient $k_X$ (Schwartz, 1986)

$$k_X = \left( \frac{r_p}{D_g} + \frac{4}{c_X \cdot \gamma_X} \right)^{-1} S_a \tag{2}$$

where $D_g$ represents the gas phase diffusion coefficient, $r_p$ is the particle radius, $c_X$ is the mean molecular velocity of the
species X, $\gamma_X$ represents the reaction probability and $S_a$ the surface area density. The $\gamma_X$ values are generally determined from

laboratory measurements. The first term in Eqn. 2 is very small for particles of accumulation mode and larger, and is neglected in the EMEP model. The main challenges for chemistry transport models are the calculation of a proper surface area density $S_a$ and the parametrization of the reaction probability $\gamma_X$.

First, Table 1 summarizes the heterogeneous reactions investigated in this study. Second, sections 3.1-3.6 discuss literature
values of $\gamma$ associated with each reaction. An overview of the parametrization and values used for the different reaction probabilities is given in Table 2. ECHAM-HAMMOZ and EMEP MSC-W use the same reaction probabilities or functions for many reaction, with the most important difference being the lack of ammonium nitrate aerosol in ECHAM-HAMMOZ. Lastly, to check if the surface area density is realistic, simulated $S_a$ is compared to a satellite-model-product in section 5.1.

## 3.1  $N_2O_5$

$N_2O_5$ reaction probability depends on aerosol water content and aerosol composition. Therefore, several laboratory studies measured $\gamma$ values on different aerosol types leading to the possibility to derive detailed parametrizations (Riemer et al., 2003b, 2009; Evans and Jacob, 2005; Liao and Seinfeld, 2005; Davis et al., 2008; Bertram and Thornton, 2009; Griffiths et al., 2009; Brown and Stutz, 2012). For dry sulphate aerosol, reaction probabilities range between $10^{-4}$ and $10^{-3}$; for wet aerosol $\gamma$ ranges between $10^{-3}$ and $8.6 \times 10^{-2}$ depending on relative humidity. The $N_2O_5$ heterogeneous reaction humidity dependence
also explains the range of reaction probabilities of sea salt aerosol of $6 \times 10^{-3}$ to $4 \times 10^{-2}$. On nitrate containing aerosol lower reaction probabilities were found due to nitrate effect (Wahner et al., 1998), between $3 \times 10^{-4}$ and $3 \times 10^{-3}$ (Chang et al. 2011, and references therein). Moreover, $N_2O_5$ can react on organic aerosol under dry conditions with low reaction probabilities in the order of $10^{-6}$ and $10^{-5}$ (Gross et al., 2009). This value increases to $10^{-4} - 10^{-3}$ under wet conditions, because the higher water content allows $N_2O_5$ to hydrolyze (Thornton et al., 2003). Even dust aerosols can be covered by a layer of water
leading to a reaction probability between $3 \times 10^{-3}$ at 30 % and $2 \times 10^{-2}$ at 70% relative humidity (Bauer et al., 2004). For $N_2O_5$ reaction on black carbon, Sander et al. (2006) reported a wide range of reaction probabilities, between $2 \times 10^{-2} - 10^{-6}$.

Most studies have used laboratory data to estimate $\gamma$ values, but some have made use of ambient data. Brown et al. (2009) used aircraft measurements over Texas, and found observation-based $\gamma$ values of ca. $5 \times 10^{-4}$–$6 \times 10^{-3}$, usually substantially lower (often a factor of 10) than values calculated using laboratory-based values. Using aircraft measurements around the
United Kingdom, Morgan et al. (2015) found rather high $\gamma$ values for $N_2O_5$, from ca. $1 \times 10^{-2}$–$3 \times 10^{-2}$, with strong dependencies on sulphate, and a clear suppression of $\gamma$ due to nitrate. They concluded that including the suppressive effect of organic aerosol in the uptake parameterisation leads to significant underprediction of the $\gamma$ values. Further, direct $N_2O_5$ measurements retrieved a highly daily variation of $\gamma_{N_2O_5}$ also explained by the nitrate effect leading to a mean value of $5.4 \times 10^{-3}$ ranging from $3 \cdot 10^{-5}$ to $2.9 \cdot 10^{-2}$ (Riedel et al., 2012). In Stone et al. (2014) and Wagner et al. (2013) in situ measurements of $N_2O_5$
were used to retrieve the reaction probability within the framework of a box model. In Stone et al. (2014) $\gamma_{N_2O_5}$ is varied over a range of values between 0 and 1, and found that values of $2 \cdot 10^{-1} - 2 \cdot 10^{-2}$ agreed best with observations. Wagner et al. (2013) retrieved the reaction probability of $N_2O_5$ using a box model driven by ambient wintertime observations. The reaction probability distribution ranges between $2 \cdot 10^{-3}$ and $1 \cdot 10^{-1}$, displaying a maximum at $2 \cdot 10^{-2}$.

It is clear from the studies mentioned above that great uncertainties surround both the magnitude and the chemical dependence of $\gamma$ values for $N_2O_5$. Even thorough evaluations such as those of Davis et al. (2008) or Chang et al. (2011) have little consideration of important components of the aerosol such as organic matter, and even such schemes seem to be inconsistent with the aircraft-observations discussed above. For our modelling studies, we have not tried to develop or use yet another scheme, but rather to make use of the $\gamma$ schemes already implemented in each model, with some small efforts at harmonisation to build similar reference schemes.

The equations used for EMEP MSC-W and ECHAM-HAMMOZ can be found in Table 2. Both models make extensive use of the parameterizations developed by Evans and Jacob (2005), with the largest difference being that EMEP includes ammonium nitrate (in fine particles) among the nitrate species. For $N_2O_5$ the uptake coefficients for sulphate, sea salt and organic aerosol are identical in the two models. For the reaction on dust, both models rely on Bauer et al. (2004) which was interpreted differently by Evans and Jacob (2005) and Liao and Seinfeld (2005). This small difference in the uptake coefficient formulation on dust does not lead to large differences in the resulting uptake coefficient.

EMEP MSC-W also modifies the $\gamma$ value for secondary inorganic aerosol to account for a nitrate inhibition effect (Wahner et al., 1998; Riemer et al., 2003a). This makes use of the $\gamma_{AN}$ factor presented in Davis et al. (2008) for ammonium nitrate, and merged with the sulphate factor in a manner reminiscent of Riemer et al. (2003b). First a sulphate mass fraction within the secondary inorganic aerosol (SIA) is calculated, $f_{SU} = m_{SO_4}/(m_{SO_4} + m_{NO_3})$, then $\gamma_{SIA}$ is calculated as given in Table 2.

Figure 1 illustrates the $\gamma$ values for sulphate aerosol from Evans and Jacob (2005) as a function of relative humidity (RH) and temperature for sulphate, and the RH dependency of $\gamma$ for nitrate from the Davis et al. (2008) formulation. The negative temperature dependence after 280K can be explained by increasing volatility with increasing temperature leading to less uptake on the aerosol. As described before, reaction probability increases with increasing water content in the aerosol due to enhanced $N_2O_5$ hydrolysis. Even at high RH, reaction probability on nitrate containing aerosol is not as high as in sulphate aerosol. Nevertheless, the very high $\gamma$ values found at high RH seem questionable, because the aerosol itself becomes saturated at high RH and these small water content changes should not have such a huge impact on the heterogeneous reaction.

No further parameterization considering organic coatings is used in either EMEP MSC-W or ECHAM-HAMMOZ due to the large uncertainties in this effect (e.g. Brown et al., 2009; Morgan et al., 2015), and the fact that ambient OM and its thermodynamic properties are so poorly understood (Hallquist et al., 2009). Further, sensitivity runs done with ECHAM-HAMMOZ have shown minor global impact of organic coatings (Stadtler, 2015). Figure S1 in the Supplement illustrates the $\gamma_{N_2O_5}$ values from the two models using the setups described in Sect. 4, showing values of around 0.01 - 0.04 over much of the globe.

## 3.2 NO₃

Hydrolysis of the nitrate radical $NO_3$ happens on various aerosol types depending on the water content. $NO_3$ heterogeneous reaction produces $HNO_3$ and OH in the aqueous particle phase and can be counted as a $NO_x$ sink (Rudich et al., 1998). Several laboratory studies shown $\gamma$ ranging between $10^{-4}$ and $10^{-3}$ (Rudich et al., 1996; Rudich et al., 2002). Jacob (2000)

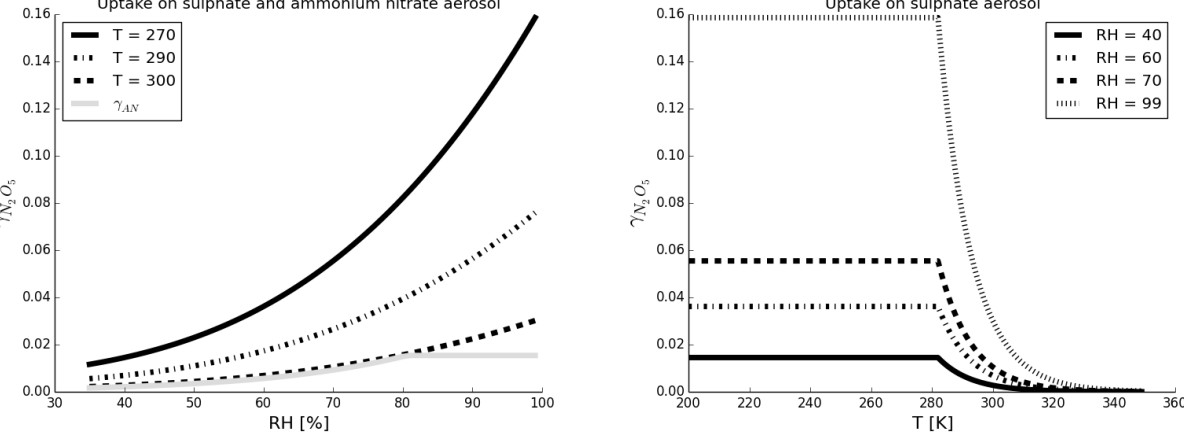

**Figure 1.** Upper plot: $N_2O_5$ reaction probability dependence on relative humidity for sulphate (black) and nitrate (grey) aerosol. For sulfate aerosol three temperatures are shown. Lower plot: $N_2O_5$ reaction probability temperature dependence on sulphate aerosol for four relative humidity. Parameterizations from Evans and Jacob (2005) and Davis et al. (2008), see text.

~~recommended to use $\gamma = 10^{-3}$ for atmospheric chemistry model simulations, and this value was adopted for EMEP and ECHAM-HAMMOZ.~~

The nitrate radical $NO_3$ undergoes hydrolysis in wet aerosols, but was also observed to react with organic compounds on the aerosol surface. Hydrolysis of nitrate radicals $NO_3$ happens on various aerossol types depending on the water content. $NO_3$

heterogeneous reaction produces $HNO_3$ and OH in the aqueous particle phase and can be counted as a $NO_x$ sink (Rudich et al., 1998). Several laboratory studies shown $\gamma$ ranging between $10^{-4}$ and $10^{-3}$ (Rudich et al., 1996; Moise et al., 2002). Jacob (2000) recommended to use $\gamma = 10^{-3}$ for atmospheric chemistry model simulations.

Reactions with different organic compounds were explored in laboratory experiments. Gross and Bertram (2008) measured the reaction probabilities between 0.059 and 0.79 of $NO_3$ with different polycyclic aromatic hydrocarbons leading to $NO_2$

and $HNO_3$ formation. Two following studies also found high reaction probabilities of $NO_3$ with alkenoic acid ($>0.07$) (Gross et al., 2009) and alkene monolayers (0.034) (Gross and Bertram, 2009). Organic coatings could enhance $NO_3$ reactive uptake, nevertheless knowledge of explicit organic compounds in the organic fraction of aerosol is unknown in both mode systems, therefore the recommended of $\gamma = 10^{-3}$ for $NO_3$ hydrolysis value was adopted for EMEP and ECHAM-HAMMOZ.

### 3.3 $NO_2$

$NO_2$ heterogeneous reaction leads to the production of $HNO_3$ and HONO. Especially in humid environments, the heterogeneous reaction may account for up to 95% of HONO production (Goodman et al., 1999). During nighttime HONO can accumulate in the atmosphere and therefore be an efficient OH radical source during the morning when sun rise starts photolysis (Goodman et al., 1999). Estimates of $\gamma$ for $NO_2$ vary widely, however, with several laboratory studies giving a range

between $10^{-8}$ and $10^{-3}$ (Harrison and Collins, 1998; Kleffmann et al., 1998; Arens et al., 2001; Underwood et al., 2001). Jacob (2000) recommended $\gamma = 10^{-4}$ and this value is used for this study.

### 3.4 HNO$_3$

Nitric acid reacts on sea salt and dust aerosol surfaces, producing nitrate which stays in the aerosol phase (Davies and Cox, 1998; Hodzic et al., 2006). Experimentally derived $\gamma$ values for HNO$_3$ on sea salt range between $10^{-4}$ and $10^{-2}$ (Davies and Cox 1998, and references therein). A relative humidity dependent uptake coefficient was proposed in Hauglustaine et al. (2014) increasing $\gamma$ from $10^{-3}$ and $10^{-1}$ to cover low and high relative humidity. No such relative humidity dependence was used in this study, because for the conditions in the marine boundary layer, the value of $10^{-2}$ fits well and is used here.

Heterogeneous reaction of HNO$_3$ on dust was studied on different types of minerals, atmospheric dust types and for a range of relative humidities giving $\gamma$ in the range of $10^{-6}$ and $10^{-1}$ (Hanisch and Crowley, 2001; Usher et al., 2003; Liu et al., 2008; Hauglustaine et al., 2014). Although in Fairlie et al. (2010) a relative humidity dependence of varying $\gamma$ between $10^{-5}$ and $10^{-3}$ is described, the value is used here is based on Hodzic et al. (2006), who tested $\gamma$ values between $10^{-6}$ and 0.3, deriving 0.1 as the best $\gamma$ value minimizing the model error compared to observations. Compared to the other referenced studies, this is an upper limit.

### 3.5 O$_3$

Studies of the heterogeneous reaction of ozone on dust give a wide range for possible reaction probabilities, from $10^{-10}$ to $10^{-4}$ (Reus et al., 2000; Usher et al., 2003; Mogili et al., 2006; George et al., 2015). Reus et al. (2000) gives $10^{-4}$ as an upper limit for this reaction probability, but George et al. (2015) suggests a more conservative upper limit value of $10^{-5}$. Nevertheless, Nicolas et al. (2009) conclude that a reaction probability of $10^{-6}$ is a realistic number in terms of atmospheric environmental conditions, and this value was adopted here.

### 3.6 HO$_2$

HO$_2$ reaction probability is highly variable and strongly depends on transition metal ions contained in the aerosol (Tilgner et al., 2005; Mao et al., 2013; George et al., 2013; Huijnen et al., 2014). Furthermore, this reaction can also take place on cloud droplets. Estimates for $\gamma$ range between 0.02 and 1 (Jacob, 2000; Remorov et al., 2002; Thornton and Abbatt, 2005; Taketani et al., 2008; George et al., 2013; Mao et al., 2013). Whalley et al. (2015) measured HO$_2$ in clouds and found a decrease in HO$_2$ concentrations up to 90%. Depending on the compounds in the particle aqueous phase heterogeneous reaction of HO$_2$ produces either H$_2$O$_2$ or H$_2$O. Consequently, this heterogeneous reaction can be a terminal radical sink or not (Mao et al., 2013; Whalley et al., 2015). Here we do not account for a terminal sink, but let the heterogeneous reaction of HO$_2$ produce H$_2$O$_2$ and use the $\gamma$ recommended by Jacob (2000) of 0.2.

## 4 Setup of sensitivity runs

To test the six heterogeneous reactions (see Table 1), six sensitivity runs were designed and performed with both models. For EMEP MSC-W a spin up of six months and for ECHAM-HAMMOZ one of twelve months is used. Afterwards the results for the whole year 2012 are evaluated. The reference run REF contains all heterogeneous reactions with the parameterizations given in Table 2. Each sensitivity run is done with five out of six heterogeneous reactions; the names of the runs show which compound does not undergo heterogeneous reaction. For example, in the noN2O5 run, only the $N_2O_5$ heterogeneous reaction is turned off. An overview of the simulations is given in Table 3.

## 5 Results and Discussion

### 5.1 Surface area density

Aerosols consist of a variety of compounds in the gas, liquid or solid phase, and the shapes of aerosols vary greatly (Pöschl, 2005). Large scale models can not explicitly treat the morphology of aerosols. In EMEP MSC-W and ECHAM-HAMMOZ distribution functions and median radii are used to simulate the aerosol population. Based on this approach surface area density $S_a$ is calculated considering the aerosol distribution, the median radius and assuming spherical particles. This assumption is good for liquid aerosols behaving as small water droplets. For dry particles this assumption can lead to an underestimation of $S_a$ due to folded or porous structures (Buseck and Posfai, 1999).

In van Donkelaar et al. (2015) satellite retrievals and the GEOS-Chem chemical transport model are used to derive global surface $PM_{2.5}$ estimates with a resolution of 10 km × 10 km in the time period between 1998 and 2012. The physical relation between AOD and surface area is described in the supplementary material of van Donkelaar et al. (2015).

Figure 2 shows the estimated $PM_{2.5}$ surface area by van Donkelaar et al. (2015) and the modeled surface area density $S_a$ from EMEP MSC-W and ECHAM-HAMMOZ as ground-level annual mean 2012 over land. Although these data-sets are not strictly comparable, since the van Donkelaar et al. 2015 estimate in itself relies partly on various assumptions of a third chemical transport model, GEOS-Chem (van Donkelaar et al., 2015), the general patterns of the models agree well with the surface area density estimation. Both models capture the east west gradient in $S_a$ over North America even if the total $S_a$ value is comparably lower in both models. Similarly, Europe in the satellite GEOS-Chem product has slightly higher $S_a$ values than the models produce. In contrast, the $S_a$ values over India are captured very well, and the peak values in East Asia are also produced by both models, while ECHAM-HAMMOZ simulates highest $S_a$ values among the three data sets in East Asia. An overestimation of both models compared to satellite GEOS-Chem happens over North Africa. In South America EMEP MSC-W performs better than ECHAM-HAMMOZ due to larger contributions from secondary organic aerosol (SOA) formation. EMEP uses a more complex SOA scheme (Bergström et al., 2012; Simpson et al., 2012) which allows for oxidation ('aging') of semivolatile organic vapours. In ECHAM-HAMMOZ an adjusted amount of organic material covering also SOA is emitted, but the amount does not close the gap leading to a lower $S_a$ compared to EMEP MSC-W and satellite GEOS-Chem.

## 5.2 Impacts of sensitivity tests

To evaluate the impact of our heterogeneous reactions, the six sensitivity runs were compared to the reference run containing all heterogeneous reactions. By turning off one heterogeneous reaction in each sensitivity run, the impact of each reaction can be estimated. Tables 4-5 show the differences in percent between the sensitivity runs and the reference run for EMEP MSC-W and ECHAM-HAMMOZ, as averaged over regions of North America (NA), Europe (EUR), East Asia (EA) and South Asia (SA) (regions defined as in Fiore et al. 2009). Tables S1-S2 in Supplementary give absolute differences, in ppb or ppt. The main focus of this evaluation is on the effect of the heterogeneous reactions on ozone mixing ratios. (Also, in order to test the importance of year to year variability, the EMEP model was additionally run for the year 2011. The results, given in Supplementary Table S3, are almost identical to those shown for 2012 in Table 4 and so not discussed further here.)

For the reference runs, EMEP MSC-W and ECHAM-HAMMOZ simulate very similar values for ozone, ECHAM-HAMMOZ giving somewhat lower mixing ratios. Also $NO_x$ values are similar in Asia, but differ by ca. 60–100% in North America and Europe. In terms of other reactive nitrogen species, EMEP MSC-W has overall higher $NO_y$ levels and especially PAN. This difference in $NO_y$ availability is expected given the impact of EMEP's $NH_3$ emissions in the formation of ammonium nitrate, thus extending the lifetime of reactive nitrogen species, on top of general differences in emissions and chemical mechanisms.

Tables 4–5 clearly show increases in $O_3$ from all the sensitivity runs when the heterogeneous reaction is turned off, except for the sensitivity run without heterogeneous $NO_2$ reaction in East Asia. In this region the special case of ozone titration (Wild and Akimoto, 2001) leads to an ozone loss due to $NO_2$ instead of production: lowering $NO_2$ in this region of very high $NO_x$ regions means reducing a loss process. Even if the models agree on the direction of the impact of heterogeneous reaction on $O_3$, they do not agree on the strength of the reactions.

For both models the $N_2O_5$ reactions have generally (ECHAM-HAMMOZ) or always (EMEP) the biggest effect on $O_3$, with changes of ca. 2–3 ppb (5–9%). Some other heterogeneous reactions (especially $NO_2$, $HO_2$ and $HNO_3$) gain some significance in highly polluted areas where aerosol surface areas are high, but the two models show quite different responses though in their response to these other gas-aerosol reactions. The EMEP model actually shows rather small impacts of all reactions on $O_3$ except $N_2O_5$, except in East and South Asia where some impacts can approach 10-20% of that of $N_2O_5$. ECHAM-HAMMOZ, on the other hand, shows quite marked responses to especially the $HNO_3$ reactions, but also the $HO_2$ reactions.

The strong response of $O_3$ in ECHAM-HAMMOZ to the $HNO_3$ reaction compared to EMEP seems to be the result of a number of factors. The simplest is that EMEP allows this reaction only on coarse aerosol, and so has a smaller surface area for this reaction, especially on dust. Another explanation is that the model sensitivities to $NO_x$ changes may be different, possibly caused by chemical differences or the different horizontal resolutions of the models. Ozone chemistry (and even the switch from production to loss) can be very sensitive to $NO_x$ concentration levels, especially in unpolluted areas (Crutzen et al., 1999; Sillman et al., 1990). $NO_x$ plumes from ships or power plants emitted into large model grid cells might well produce more $O_3$ in one model than the other, leading to different sensitivities to $NO_x$ emissions (von Glasow et al., 2003; Vinken et al., 2011). The EMEP model has in fact a psuedo-species 'SHIPNOx' by which 50% of $NO_x$ from ship plumes are given a

pathway to $HNO_3$ production, skipping the intermediate $NO_2$ production associated with overestimating $O_3$ production from NO in pristine environments (Simpson et al., 2015). A further factor is the lack of nitrate aerosol in ECHAM-HAMMOZ. In the EMEP model $HNO_3$ can take part in ammonium nitrate aerosol (AN) formation, thus extending the lifetime of $NO_y$. Due to the AN, some $HNO_3$ can be recycled back into the atmosphere stabilizing the $HNO_3$ and $NO_3$ mixing ratios.

Table S2 (and 5) shows that for ECHAM-HAMMOZ omitting the $HNO_3$ reaction on dust and sea salt aerosol increases $NO_x$ by ca. 10-20 ppt (1 %), whereas in EMEP the change is tiny ($\leq$ 1 ppt). The impact in ECHAM-HAMMOZ can be found over the whole globe, but especially over the oceans, where $NO_x$ is low, but still much higher than $NO_z$ (=$NO_3$+$N_2O_5$). Changes in $NO_z$ with this noHNO3 scenario are far higher in ECHAM-HAMMOZ than in EMEP. Even if heterogeneous $HNO_3$ loss does not hugely impact $NO_x$, a small $NO_x$ increase, even if really small, is ubiquitous and can shift the equilibrium between
ozone production and loss towards more production, reaching a higher steady state $O_3$ concentration. Also, this reaction has a significant effect on $NO_3$, reducing it in the northern oceans by about 10 % (not shown). ~~$NO_3$ rapidly photolyses, and resulting $NO_2$ likewise, so has a high ozone-formation potential.~~ $NO_3$ rapidly photolyses and produces $NO_2$ and atomic oxygen $O_3(^3P)$. $NO_2$ subsequently photolyses and results in NO and a second $O_3(^3P)$. From these two reactions two ozone molecules can be formed, therefore $NO_3$ has a high ozone-formation potential. Reducing $HNO_3$ and therefore $NO_3$ drastically by the
surface reaction in this highly sensitive region leads to a nonlinear response of the model changing the gross ozone production in ECHAM-HAMMOZ by 350 Tg, which is a reduction of 7 %. This leads to a global more or less uniformly distributed difference of 1 - 4 ppbv in ozone mixing ratios.

Analyzing all the possible differences in these two different models is beyond of the scope of this study, but it may well be that ECHAM-HAMMOZ overestimates the impact of $HNO_3$ due to missing nitrate aerosol formation and EMEP underestimates
the impact, due to the use of only coarse sea salt and dust aerosol for the $HNO_3$ and $HO_2$ reactions.

As the $N_2O_5$ reactions have the greatest impact on tracer concentrations among our sensitivity tests, the spatial and temporal differences between the reference run and the sensitivity run noN2O5 have been investigated in more detail. Figs. 3 and 4 show the difference between the mixing ratios of $O_3$ and $NO_x$ in the reference run and in the sensitivity run without the $N_2O_5$ reactions. Both models show the largest changes in regions where high aerosol loadings and high $NO_x$ emissions can be found,
such as Northeast America, Europe, South and East Asia.

Converting $N_2O_5$ to $HNO_3$ on aerosol surfaces introduces an additional sink for $NO_x$, because $HNO_3$ is rapidly (in EMEP) or immediately (in ECHAM-HAMMOZ) lost via dry and wet deposition, and reactive uptake on aerosols, after it is produced. Therefore, $NO_x$ mixing ratios are lowered in the reference run REF compared to the simulation without the heterogeneous reaction noN2O5, as can be seen in Figure 4.

For ozone, the differences propagate through the whole northern hemisphere due to the longer lifetime of $O_3$ compared to $NO_x$ (Fig. 3). Again both models simulate similar patterns with regard to the spatial distribution of changes due to $N_2O_5$. ~~As described before, EMEP MSC-W shows a bigger impact on $O_3$ concentrations than ECHAM-HAMMOZ because of $O_3$ is already lowered due to the direct heterogeneous loss of $HNO_3$.~~

Especially for East Asia the impact of heterogeneous reactions cannot be neglected. High nitrate loadings in ammonium
poor regions verify the importance shown by other models (Pathak et al., 2008). In the southern hemisphere, $N_2O_5$ and the

other heterogeneous reactions evaluated in this study have much smaller impacts on ozone and $NO_x$ than seen in the northern hemisphere (Figures 3, 4).

To explore the seasonal impact of $N_2O_5$ reactions, Fig. 5 shows monthly values for tracer mixing ratios and surface area density from both models for the different northern hemispheric regions. In general, the models produce comparable seasonal cycles for the gas tracers and surface area density. Strongest changes in seasonal cycles are found in the noN2O5 run. ~~and in ozone for both runs~~. In the noN2O5 run, $N_2O_5$ builds up during winter time, because it is thermally unstable and photolabile. Including the heterogeneous uptake leads to a strong $N_2O_5$ reduction in both models, yielding a flatter seasonal curve. The loss of $N_2O_5$ in going from noN2O5 to REF leads to a decrease in $NO_2$, $NO_3$ and PAN. Here models slightly differ. EMEP displays a stronger reduction in $NO_3$ and PAN, since it has in both runs higher mixing ratios compared to ECHAM-HAMMOZ. Removing $NO_2$ from the system leads in both models to a reduction of ozone. Although, the impact of $N_2O_5$ heterogeneous reaction on $NO_2$ is higher in winter and lowest during summer, the greatest change in $O_3$ can be found during spring. This can be explained by the availability of $N_2O_5$ and ozone production strength. As stated before, $N_2O_5$ is formed during night time, therefore less sun is favorable. In contrast, to form ozone light is needed. Still high $N_2O_5$ concentrations, enough surface area and a sufficiently high ozone production can be found during spring, leading to the biggest change in $O_3$ production during this season. ~~During winter, nights are longer, the sun is less active, therefore heterogeneous chemistry is efficient, but less ozone production reduction occurs.~~ During winter, nights are longer leading to inactive photochemistry. Therefore, heterogeneous chemistry is efficient. Nevertheless, a rather inactive photochemistry also leads to less ozone production. Comparing to spring, the impact seen here is lower because of already low ozone formation rate.

**Table 1.** Heterogeneous reactions in the EMEP MSC-W and ECHAM-HAMMOZ models. The second column specifies the aerosol type on which the reaction proceeds in the models:. SS: seasalt, DU: dust, PM: particulate matter

| No. | Reaction | | Aerosol type | Notes |
|-----|----------|--|--------------|-------|
| R1 | $N_2O_5$ | $\rightarrow 2HNO_3$ | PM | [1] |
| R2 | $NO_3$ | $\rightarrow HNO_3$ | PM | [2] |
| R3 | $NO_2$ | $\rightarrow 1/2HNO_3 + 1/2HONO$ | PM | [2] |
| R4 | $HNO_3$ | $\rightarrow NO_3^-$ | SS, DU | [2,3] |
| R5 | $HO_2$ | $\rightarrow 1/2H_2O_2$ | PM | [2] |
| R6 | $O_3$ | $\rightarrow HO_2$ | DU | |

[1] Just for RH>40% in EMEP.

[2] Just on wet aerosol in ECHAM-HAMMOZ.

[3] Just on coarse mode dust and sea-salt in EMEP, using $S_{ss}$ and/or $S_{du}$, see Sect. 2.1.

**Table 2.** Reaction probabilities for the different species. Unless explicitly labelled (in parentheses afer the equation), both models use the same formulation. Here, RH denotes relative humidity in range, $RH \in (0, 100)$, and fRH denotes fractional relative humidity in range, $fRH \in (0, 1)$.

| Specie | $\gamma$ | Reference |
|---|---|---|
| $N_2O_5$ | $\gamma_{SS} = \begin{cases} 0.005, RH \leq 62\ \% \\ 0.03, RH \geq 62\ \% \end{cases}$ | EVA05 |
| | $\gamma_{SU} = \alpha \cdot 10^{-\beta}$ | EVA05 |
| | $\alpha = 2.79 \cdot 10^{-4} fRH + 1.3 \cdot 10^{-4} fRH - 3.43 \cdot 10^{-6} fRH^2 + 7.52 \cdot 10^{-8} fRH^3$ | |
| | $\beta = \begin{cases} 4 \cdot 10^{-2}(T - 294), T > 282\ K \\ -0.48, T \leq 282\ K \end{cases}$ | |
| | $\gamma_{DU} = 0.01$ (EMEP) | EVA05 |
| | $\gamma_{DU} = 4.25 \cdot 10^{-4} RH - 9.75 \cdot 10^{-3} (30\% \leq RH \leq 70\%)$ (ECHAM) | LIA05 |
| | $\gamma_{OC} = \begin{cases} 0.03, RH > 57\% \\ 5.2 \cdot 10^{-2}, RH \leq 57\% \end{cases}$ | |
| | $\gamma_{BC} = 0.005$ | EVA05 |
| | $\gamma_{AN} = \min(0.0154, 1/(1 + \exp(8.10774 - 0.04902 \cdot RH)))$ (EMEP) | DAV08 |
| | $\gamma_{SIA} = f_{SU} \gamma_{SU} + (1 - f_{SU}) \gamma_{AN}$ (EMEP) | SIM15 |
| | (where $f_{SU}$ = mass fraction of sulfate in inorganic aerosol, see Sect. 3.1) | |
| $NO_3$ | $\gamma = 0.001$ | JAC00 |
| $NO_2$ | $\gamma = 10^{-4}$ | JAC00 |
| $HNO_3$ | $\gamma_{SS} = 0.01$ | DAV98 |
| | $\gamma_{DU} = 0.1$ | HOD06 |
| $HO_2$ | $\gamma = 0.2$ | JAC00 |
| $O_3$ | $\gamma_{DU} = 10^{-6}$ | NIC09 |

The subscripts refer to the aerosol compounds as given in Table 1, plus OC:Organic carbon/Organic matter, SU: Sulphate, SIA:secondary inorganic aerosol, BC black carbon.

Refs: DAV98 Davies and Cox (1998), DAV08 Davis et al. (2008), EVA05 Evans and Jacob (2005), JAC00 Jacob (2000), LIA05 Liao and Seinfeld (2005), THO08 Thornton et al. (2008), HOD06 Hodzic et al. (2006), NIC09 Nicolas et al. (2009) SIM15 Simpson et al. (2015)

**Table 3.** Overview of sensitivity runs.

| Run | Description |
| --- | --- |
| REF | All heterogeneous reactions |
| noN2O5 | All except $N_2O_5$ reaction |
| noNO3 | All except $NO_3$ reaction |
| noNO2 | All except $NO_2$ reaction |
| noHNO3 | All except $HNO_3$ reaction |
| noHO2 | All except $HO_2$ reaction |
| noO3 | All except $O_3$ reaction |

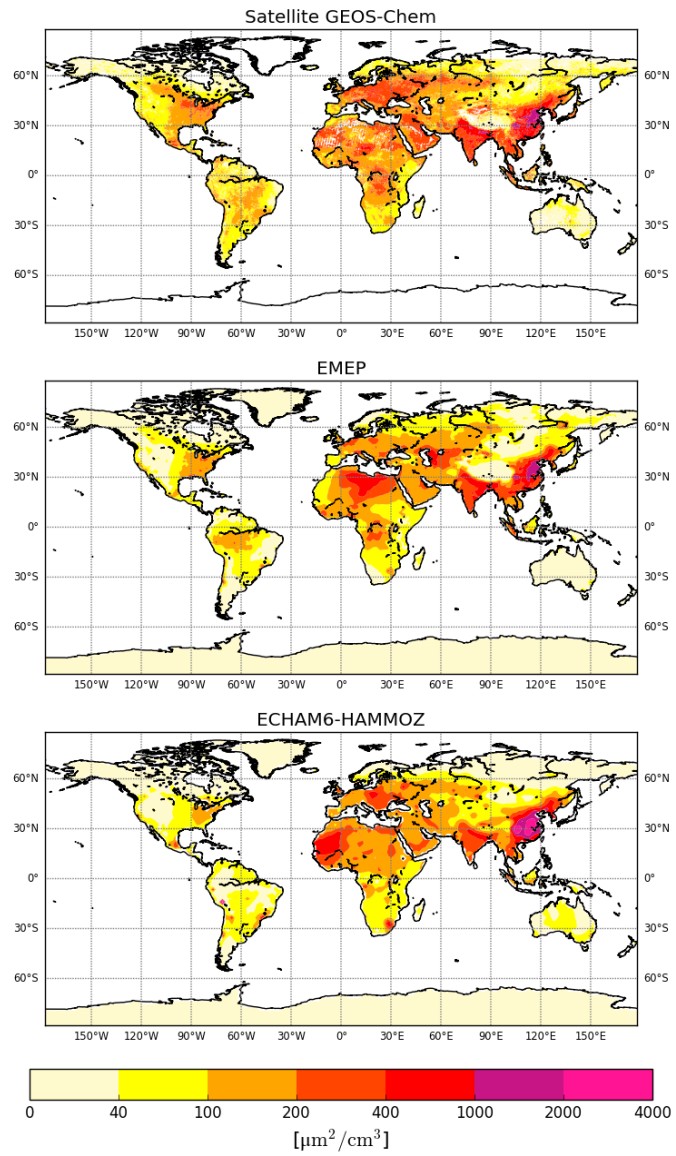

**Figure 2.** Satellite estimated (top) and simulated surface area densities by EMEP (middle) and ECHAM-HAMMOZ (bottom) at ground level. The Satellite data is an average value for the time period 2010 - 2012, from van Donkelaar et al. (2015). The model data is for 2012 and the lowest model level.

**Table 4.** Impacts of gas-aerosol reactions on regional ground level average mixing ratios of $O_3$ and key $NO_y$ compounds: EMEP model, year 2012.

| Region | Run | Unit | $O_3$ | $NO_x$ | $NO_y$ | $HNO_3$ | PAN | $N_2O_5$ | $NO_3$ |
|--------|-----|------|-------|--------|--------|---------|-----|----------|--------|
| NA | REF | Conc*: | 40.33 | 0.82 | 1.81 | 0.21 | 0.55 | 5.08 | 4.54 |
| NA | noN2O5 | %: | 5 | 9 | 4 | -10 | 8 | 160 | 59 |
| NA | noHO2 | %: | 0 | -1 | 0 | 1 | 1 | -2 | -10 |
| NA | noHNO3 | %: | 0 | 0 | -2 | 18 | 0 | 0 | 0 |
| NA | noNO2 | %: | 0 | 2 | 1 | -1 | 0 | 4 | 2 |
| NA | noNO3 | %: | 0 | 0 | 0 | 0 | 0 | 1 | 3 |
| NA | noO3 | %: | 0 | 0 | 0 | 0 | 0 | 0 | 0 |
| | | | | | | | | | |
| EUR | REF | Conc*: | 40.89 | 1.01 | 2.43 | 0.25 | 0.54 | 7.73 | 6.48 |
| EUR | noN2O5 | %: | 7 | 16 | 3 | -16 | 10 | 280 | 72 |
| EUR | noHO2 | %: | 1 | -3 | 0 | 1 | 4 | -4 | -14 |
| EUR | noHNO3 | %: | 1 | 0 | -6 | 58 | 0 | 0 | 3 |
| EUR | noNO2 | %: | 0 | 5 | 1 | -1 | -1 | 6 | 4 |
| EUR | noNO3 | %: | 0 | 0 | 0 | 0 | 0 | 2 | 10 |
| EUR | noO3 | %: | 0 | 0 | 0 | 0 | 0 | 0 | 2 |
| | | | | | | | | | |
| EA | REF | Conc*: | 43.96 | 2.23 | 4.63 | 0.54 | 0.89 | 12.59 | 5.52 |
| EA | noN2O5 | %: | 8 | 14 | 4 | -19 | 13 | 278 | 106 |
| EA | noHO2 | %: | 2 | -4 | 0 | 2 | 7 | 0 | -7 |
| EA | noHNO3 | %: | 0 | 0 | -2 | 13 | 0 | 0 | 0 |
| EA | noNO2 | %: | -1 | 30 | 9 | -11 | -8 | 13 | 4 |
| EA | noNO3 | %: | 0 | 0 | 0 | 0 | 0 | 1 | 3 |
| EA | noO3 | %: | 0 | 0 | 0 | 0 | 0 | 0 | 0 |
| | | | | | | | | | |
| SA | REF | Conc*: | 47.33 | 1.12 | 2.90 | 0.42 | 0.33 | 10.37 | 12.04 |
| SA | noN2O5 | %: | 6 | 11 | 1 | -4 | 15 | 139 | 63 |
| SA | noHO2 | %: | 1 | -3 | 0 | 1 | 5 | -5 | -12 |
| SA | noHNO3 | %: | 1 | 0 | -8 | 61 | 0 | 1 | 4 |
| SA | noNO2 | %: | 1 | 4 | 1 | 0 | 1 | 10 | 5 |
| SA | noNO3 | %: | 1 | 0 | 0 | 0 | 1 | 5 | 11 |
| SA | noO3 | %: | 0 | 0 | 0 | 0 | 0 | 0 | 0 |

Notes: Base-case concentrations from the surface-level of the model are given in ppt for $NO_3$ and $N_2O_5$, otherwise ppb (Conc* flags this difference in units). Results for the sensitivity tests are given as (test-base)/base in %. The first column refers to the region over which the annual mean is spatially averaged, and the second column refers to the corresponding run. Regions are defined as follows: NA ($15°$N–$55°$N; $60°$W–$125°$W), EU ($25°$N–$65°$N; $10°$W-$50°$E), EA ($15°$N–$50°$N; $95°$E–$160°$E), and SA ($5°$N–$35°$N; $50°$E–$95°$E).

**Table 5.** Impacts of gas-aerosol reactions on regional ground level average mixing ratios of $O_3$ and key $NO_y$ compounds. As Table 4, but for the ECHAM-HAMMOZ model.

| Region | Run | Unit | $O_3$ | $NO_x$ | $NO_y$ | $HNO_3$ | PAN | $N_2O_5$ | $NO_3$ |
|--------|-----|------|-------|--------|--------|---------|-----|----------|--------|
| NA | REF | Conc* | 38.94 | 1.29 | 1.59 | 0.15 | 0.14 | 14.85 | 2.71 |
| NA | noN2O5 | % | 6 | 7 | 8 | 8 | 8 | 94 | 56 |
| NA | noHO2 | % | 0 | -1 | -1 | 2 | 1 | 0 | -3 |
| NA | noHNO3 | % | 6 | -1 | 11 | 127 | 2 | 6 | 11 |
| NA | noNO2 | % | 0 | 2 | 2 | -1 | -1 | 3 | 2 |
| NA | noNO3 | % | 0 | 0 | 0 | 0 | 0 | 0 | 1 |
| NA | noO3 | % | 0 | 0 | 0 | 0 | 0 | 0 | 0 |
| | | | | | | | | | |
| EUR | REF | Conc* | 39.57 | 2.03 | 2.38 | 0.15 | 0.16 | 21.51 | 4.7 |
| EUR | noN2O5 | % | 7 | 11 | 13 | 12 | 14 | 177 | 61 |
| EUR | noHO2 | % | 1 | -2 | -1 | 3 | 6 | -1 | -4 |
| EUR | noHNO3 | % | 5 | -1 | 14 | 227 | 2 | 6 | 14 |
| EUR | noNO2 | % | 0 | 6 | 4 | -3 | -5 | 5 | 3 |
| EUR | noNO3 | % | 0 | 0 | 0 | 0 | 0 | 1 | 3 |
| EUR | noO3 | % | 0 | 0 | 0 | 0 | 0 | 0 | 0 |
| | | | | | | | | | |
| EA | REF | Conc* | 38.51 | 2.1 | 2.54 | 0.17 | 0.26 | 10.05 | 2.64 |
| EA | noN2O5 | % | 9 | 11 | 13 | 15 | 15 | 311 | 114 |
| EA | noHO2 | % | 2 | -5 | -2 | 5 | 13 | 1 | -2 |
| EA | noHNO3 | % | 5 | -1 | 10 | 148 | 1 | 5 | 9 |
| EA | noNO2 | % | 0 | 29 | 21 | -8 | -21 | 10 | 5 |
| EA | noNO3 | % | 0 | 0 | 0 | 0 | 0 | 0 | 2 |
| EA | noO3 | % | 0 | 0 | 0 | 0 | 0 | 0 | 0 |
| | | | | | | | | | |
| SA | REF | Conc* | 44.26 | 1.29 | 1.53 | 0.1 | 0.12 | 15.5 | 6.15 |
| SA | noN2O5 | % | 5 | 6 | 8 | 13 | 14 | 96 | 35 |
| SA | noHO2 | % | 1 | -3 | -2 | 5 | 6 | -1 | -4 |
| SA | noHNO3 | % | 8 | -1 | 39 | 612 | 4 | 8 | 17 |
| SA | noNO2 | % | 1 | 4 | 3 | -1 | 0 | 8 | 5 |
| SA | noNO3 | % | 0 | 0 | 0 | 0 | 0 | 2 | 3 |
| SA | noO3 | % | 0 | 0 | 0 | 0 | 0 | 0 | 0 |

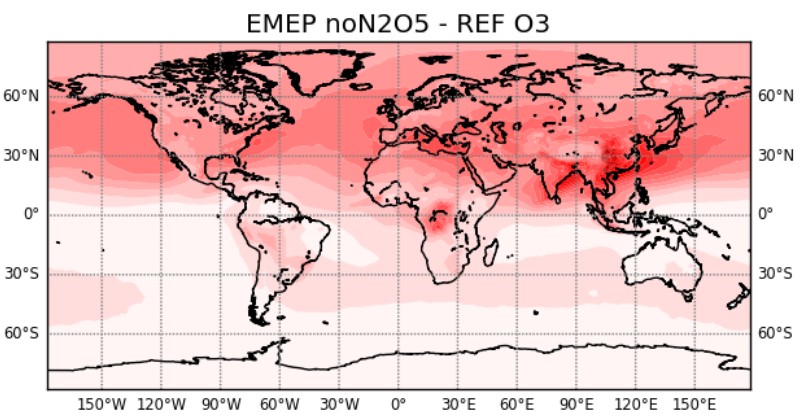

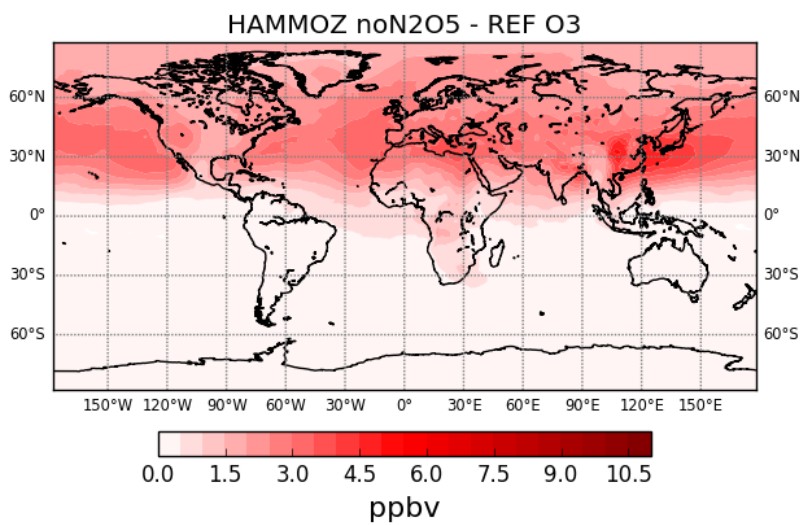

**Figure 3.** Differences in annual mean ground level ozone mixing ratio between the reference run REF and the sensitivity run noN2O5 for 2012. Since the Reference run was subtracted from the noN2O5 run, positive values show higher values in noN2O5 than in REF.

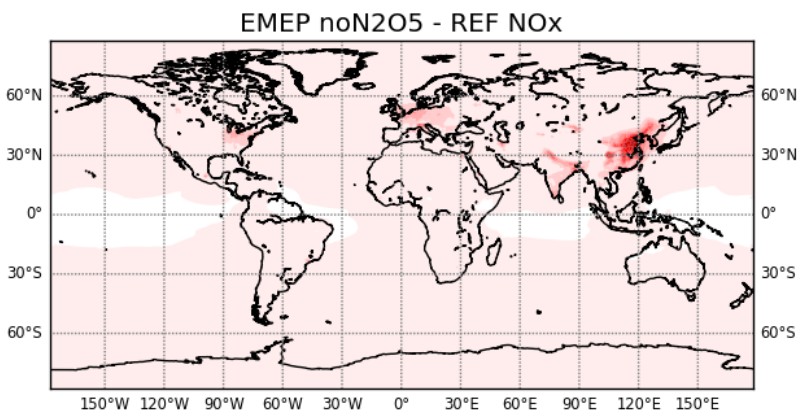

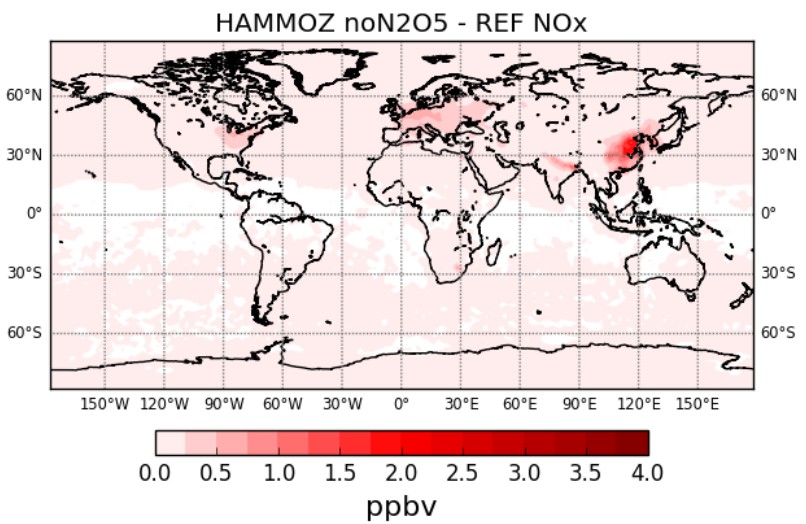

**Figure 4.** As Fig. 3, but for annual mean ground level NO$_x$ mixing ratios.

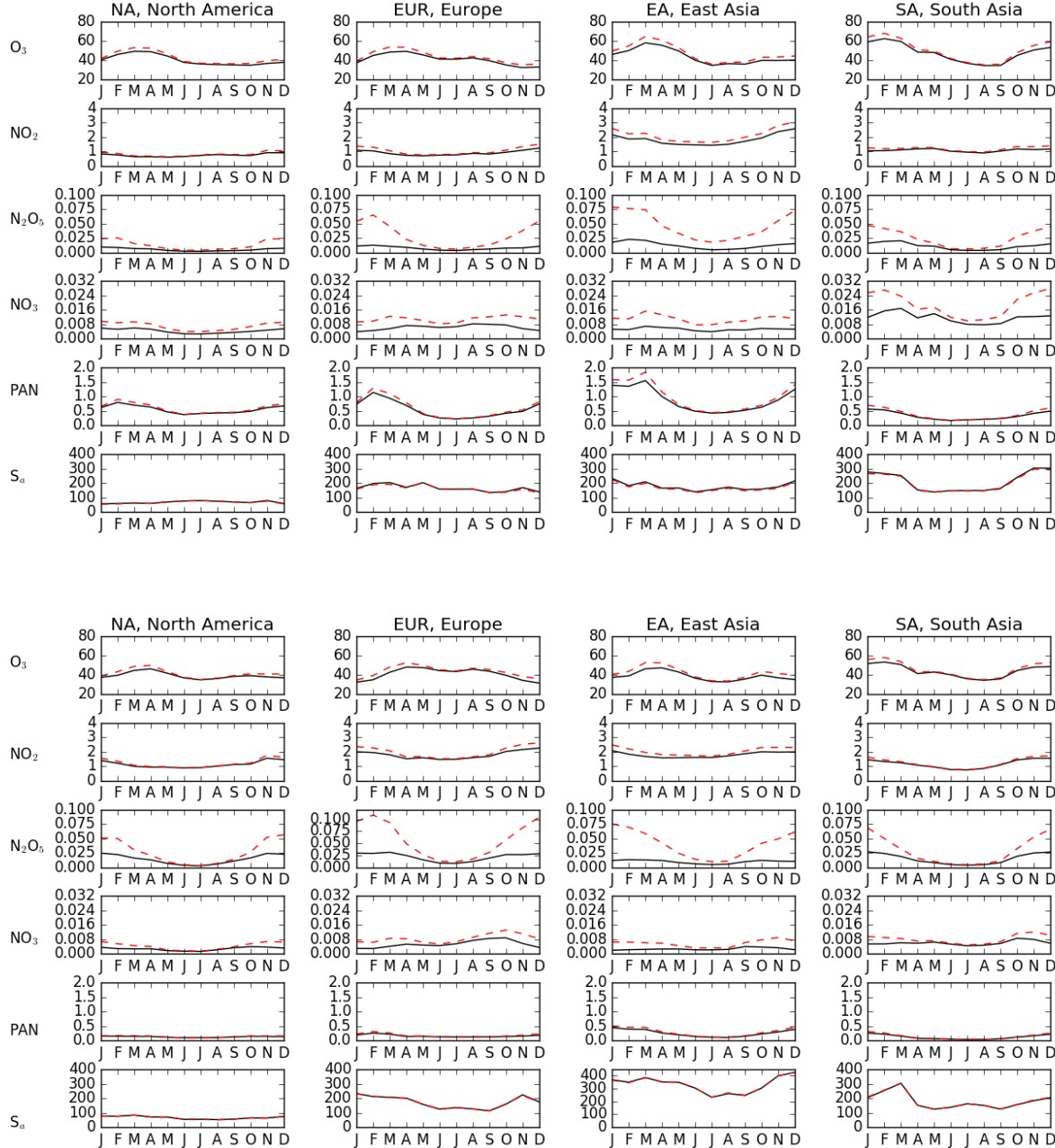

**Figure 5.** Changes in near-surface $O_3$, $NO_2$, $N_2O_5$, $NO_3$, PAN and $S_a$ for the base-case (solid black line) and noN2O5 case (dashed red line) for EMEP (top) and ECHAM-HAMMOZ (bottom). Plots show monthly gas phase mixing ratios in ppbv and surface area density in $\mu m^2 cm^{-3}$ for different regions as defined above.

An important question is how sensitive the results are to the particular values chosen for the $\gamma$ values. This is a complicated question, since these reactions also change the composition of $NO_y$ in the atmosphere, the lifetime of $NO_2$ and hence the photooxidation processes leading to $O_3$. In order to address this, additional runs with the EMEP model in four new configurations were performed:

1. $\gamma = 0.01$ for $N_2O_5$, a value lower than typical values, and at the low end of estimates (see Section 3.1).

2. $\gamma = 0.1$ for $N_2O_5$, equivalent to values used by e.g. Dentener and Crutzen (1993); Tie et al. (2001, 2003), which is substantially higher than values obtained for $\gamma_{N_2O_5}$ used here (Table 2, Figure S1).

3. $\gamma = 1.0 \times 10^{-3}$ for $NO_2$, at the top end of estimates (Section 3.3).

4. $\gamma = 0.0$ for $NO_2$, since the lowest estimates are extremely low.

The model has been run for new base-cases including $\gamma$ as listed above, and for the noN2O5, noHNO3 and (except for test (4)) noNO2 cases. Results for the regional averages (equivalent to manuscript Tables 4-5) are shown in Tables S4 and S5 in the supplement. Considering the $N_2O_5$ tests first, the changes in ozone over for example North America range from 3% ($\gamma = 0.01$) to 8% ($\gamma = 0.1$), compared to the original estimate in REF of 5% (Table 4). Changes for $NO_x$ follow a similar pattern (e.g. 6-13% for NA, versus original 9%), but changes for $N_2O_5$ itself are much more significant (80% versus 354%, compared to the original 160%).

Considering the $\gamma$ tests for $NO_2$, the test results for the noN2O5 tests generally span those of the original runs, e.g. changes of 4–6% for ozone in North America versus 5% in the original run, or 113–170% for $N_2O_5$ versus 160% for the original case. Test (3), with the high $\gamma = 1.0 \times 10^{-3}$ for $NO_2$ does have significant impacts on the $NO_x$ levels though, from e.g. 2% in the original run to 16% in test (3) for NA, or from 30% to 109% in East Asia. In these runs the impacts of noNO2 on ozone become comparable to those of noN2O5, and in South Asia the ozone changes from noNO2 actually exceed those from noN2O5.

Test (4), using zero $\gamma$ actually gives results which are very similar to our default $\gamma = 1.0 \times 10^{-4}$, suggesting that this reaction only becomes important if higher values than $\times 10^{-4}$ can be justified.

Thus, we find that the exact changes in ozone and N-compounds do depend on the assumed $\gamma$ values, but the relative importance of the different heterogeneous reactions generally remains. The $N_2O_5$ reactions are in nearly all cases the most important driver of ozone changes, but the use of a very high values for $\gamma$ for $NO_2$ changes the picture somewhat. We can note though that use of the high 0.1 values for $\gamma(NO_2)$ leads to quite significant reductions in annual $NO_2$ concentrations, resulting in degraded performance of the EMEP model compared to measurements (not shown), at least across the EMEP observational network in Europe (Tørseth et al., 2012).

## 5.3 Comparison with observations

Surface observations from 20 sites of the GAW and TOAR networks (Global Atmospheric Watch, Schultz et al. 2015, 2017), with stations distributed over the world, were used to evaluate ozone concentrations in the reference and $N_2O_5$ sensitivity runs of both models. The GAW data set consists of many sites in North America and Europe, but unfortunately few in Asia (e.g.

none in China for 2012). Still, sites exist in Japan and these should provide a good indication of ozone formation downwind of mainland China. Mountain sites were excluded from this comparison in order to avoid problems with the interpretation of which model level is most appropriate for comparison. Trinidad Head on the west coast of USA and Mace Head on the west coast of Ireland are also good background stations which capture trends in hemispheric air masses arriving from the Pacific and Atlantic respectively (Parrish et al., 2009, 2014). To capture the seasonal dependence of $N_2O_5$ uptake on aerosol, daily maximum ozone values were compared with the corresponding interpolated model data. Since the stations were selected to be relatively remote and low-elevation ground stations, the comparison with the coarse grids of the models might be representative.

Six out of the twenty stations are shown in Figs. 6, 7 and 8. Both models generally capture the seasonal variation well, fine structures and fluctuations are often reproduced, but not equally well by both models and depending on the station. For example in Tsukuba, Japan both models simulate the increasing variability during summer time, nevertheless peak concentrations are still underestimated. EMEP calculates higher peak values, than ECHAM-HAMMOZ, in contrast in Waldhof, Germany, ECHAM-HAMMOZ simulates higher peak values, partially overestimating them compared to the observations.

A closer look at the dashed line compared to the solid line reveals the seasonal highest impact of $N_2O_5$ during spring time. The high impact in spring pattern can be found in both model simulations , but is stronger in EMEP, since ECHAM-HAMMOZ includes the year through ozone reduction due to direct $HNO_3$ loss explained in the previous section 5.2. For example in Mace Head, Ireland the springtime ozone formation is clearly decreased by $N_2O_5$ reaction, while during summer the impact is marginal and increases again during winter. Both models start with a spin-up from the reference run, therefore the winter impact can not be seen in January. If the models would run for another month, this would show too, indicated by the gap between reference run and noN2O5 sensitivity run at the very end of the year.

Concluding, the impact of $N_2O_5$ heterogeneous reactions on chemical ozone production leads to a better agreement of EMEP and ECHAM-HAMMOZ and daily maximum ozone station observations in remote stations. Both models show improvements in model bias, which is expected with due to prior slight ozone overestimation; heterogeneous chemistry removes ozone, hence  in models which tend to overestimate it, logically inclusion of such reactions tends to improve the model performance with regard to bias. Such improvements could also arise if introducing other nitrogen species loss processes to reduce ozone production, having less $NO_x$ emissions or dynamically inhibiting downward transport of stratospheric ozone. Especially the stratospheric ozone intrusion is assumed to strongly happen during spring time, which would cause the same pattern as we see here in Figures 6, 7 and 8.

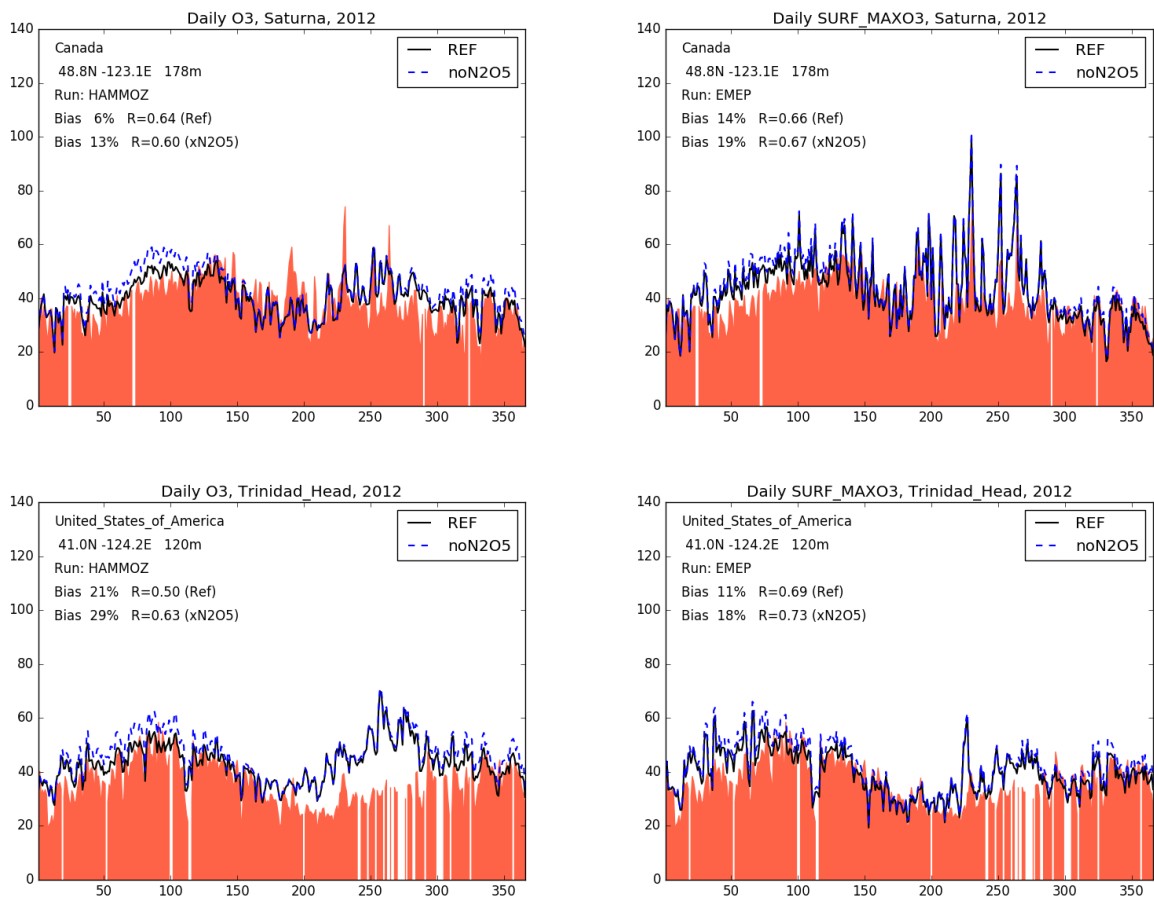

**Figure 6.** Modelled versus observed daily maximum ozone (ppbv) for two North American sites (Saturna, Canada, Trinidad Head, USA). The shaded area refers to surface station observations, the solid line is the reference run of the model and the dashed line the sensitivity run noN2O5 excluding heterogenous $N_2O_5$ reaction. On the upper left corner the station location, model, bias and correlation R are specified.

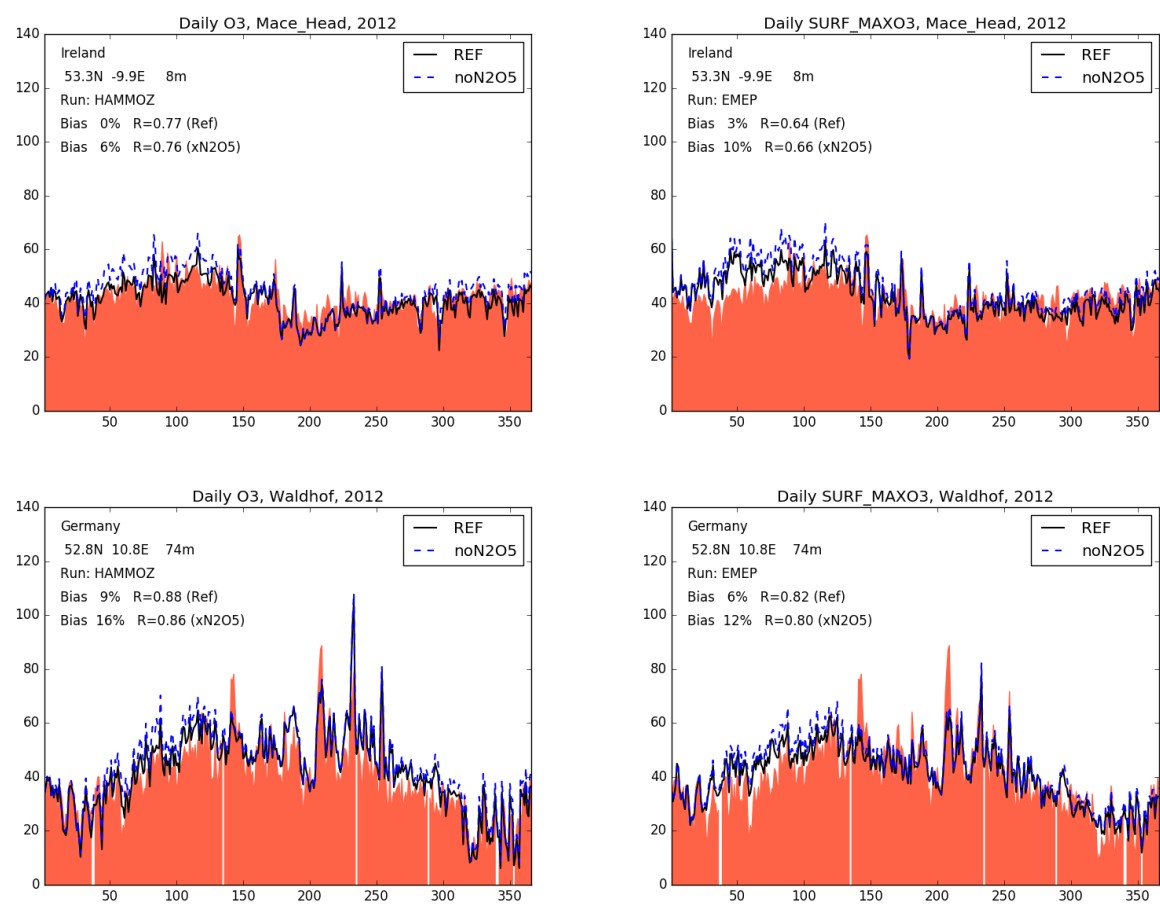

**Figure 7.** As Fig. 6 but for two European sites, Mace Head (Ireland) and Waldhof (Germany).

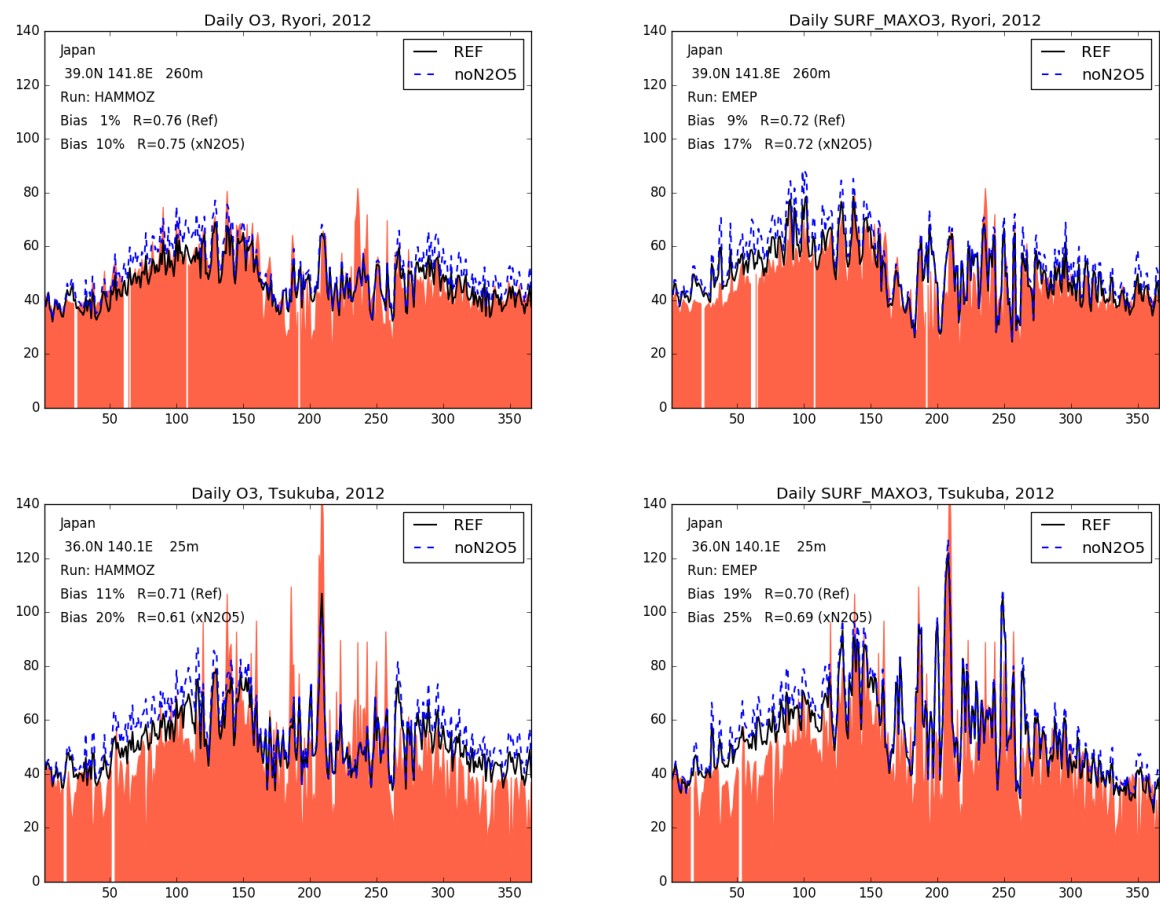

**Figure 8.** As Fig. 6 but for two Japanese sites, Ryori and Tsukuba.

## 5.4 Discussion

The influence of heterogeneous chemistry is known to be important in global chemical transport models, but it is also generally difficult to parameterize for many reasons (Jacob, 2000; Chang et al., 2011; Brown and Stutz, 2012). These include the difficulty of accurately simulating aerosol surface area density available for heterogeneous reactions and the large uncertainty in uptake coefficients. In this section we briefly compare our results with some previous studies, and also comment on some of the remaining difficulties which will need to be tackled in future studies. Concerning modelling, many studies have been published on especially the importance of the $N_2O_5$ reactions, (e.g. Dentener and Crutzen, 1993; Tie et al., 2001, 2003; Evans and Jacob, 2005; Alexander et al., 2009; Macintyre and Evans, 2010; Chang et al., 2011).

Table 6 presents a comparison of some reductions in $O_3$ and $NO_x$ due to $N_2O_5$ aerosol uptake. Starting with the annual values, the classic study of Dentener and Crutzen (1993) produced the most dramatic changes in $O_3$ and especially $NO_x$ (40-49%, depending on assumed $\gamma$ value), with subsequent studies including ours producing smaller changes. Macintyre and Evans (2010) explored runs with a variety of $\gamma$ coefficients, showing how $O_3$ and $NO_x$ sensitivities change with different values. Our global results can be see to lie not too far from the Macintyre and Evans (2010) results obtained with $\gamma = 0.01$.

As seen in Sect. 5.2, the impact of $N_2O_5$ hydrolysis is higher during winter in our study, and Table 6 confirms this for other studies. Dentener and Crutzen (1993) report about a 75 % $NO_x$ and 20 % $O_3$ reduction in their winter period with $\gamma = 0.1$. Although Tie et al. (2001) found such dramatic $NO_x$ changes at 45°N, reductions were much smaller elsewhere (e.g. 3% at the equator). A follow-up study of Tie et al. (2003) gave global average $NO_x$ and $O_3$ reductions of 38% and 6%, respectively, significantly lower than that found by Dentener and Crutzen (1993). Our models produce smaller changes again, for example ECHAM-HAMMOZ simulates a reduction in tropospheric $NO_x$ due to $N_2O_5$ hydrolysis of 9% in winter, (reductions in surface-level concentrations are greater, at 16 %). Also, $O_3$ reductions with our models are somewhat lower compared to these other models. EMEP shows the lowest changes in wintertime $NO_x$, though $O_3$ changes are closer to those of ECHAM-HAMMOZ.

Summertime results from Table 6 will not be discussed in detail, but again we see the same trend of more recent models producing smaller changes.

There are many possible reasons for these differences. Firstly, the $\gamma_{N_2O_5}$ values used by Dentener and Crutzen 1993 and Tie et al. (2001, 2003) (= 0.1) are significantly larger than the typical values of around 0.01-0.04 as calculated in this study (c.f. Fig. S1) and seen in atmospheric observations (Brown et al., 2009; Brown and Stutz, 2012). Macintyre and Evans (2010) tested the model sensitivity to uniform $\gamma_{N_2O_5}$ values and report the highest sensitivity between 0.001 and 0.02. This is exactly the range of values given frequently by the $\gamma_{N_2O_5}$ parametrization used here. The impact of the hydrolysis reaction on ozone is indeed stronger with higher $\gamma$, but our main results are relatively insensitive to these necessarily very uncertain choices (Sect. 5.2, Table S4). The ECHAM-HAMMOZ and EMEP models also have a set of other heterogeneous reactions competing with $N_2O_5$ hydrolysis, which again lowers the possible impact of this hydrolysis reaction.

There have been many changes in models, emissions, and indeed the atmosphere since these early studies. For example, the pioneering study of Dentener and Crutzen (1993) used had a model with a horizontal resolution of $10 \times 10°$, giving grid cells

**Table 6.** Comparison of impacts found by turning off $N_2O_5$ hydrolysis in global model studies. Reductions in $NO_x$ and $O_3$ found in the different global model studies are given in %.

| Model | domain | $\gamma_{N_2O_5}$ | $NO_x$[a] | $O_3$ | Comments |
|---|---|---|---|---|---|
| **Annual** | | | | | |
| Den93 | Trop. | 0.1 | 49[a] | 9 | globe |
| Den93 | Trop. | 0.01 | 40[a] | 4 | globe |
| ME10 | Trop | 0.01 | ∼12 | ∼2.5 | globe |
| ME10 | Trop | 0.01 | ∼30 | ∼6 | N. Extra Trop. |
| ME10 | Trop | 0.1 | ∼20 | ∼7 | globe |
| ME10 | Trop | 0.1 | ∼38 | ∼12 | N. Extra Trop. |
| HAMMOZ | Trop. | Table 2 | 9.1 | 2.0 | globe, this study |
| HAMMOZ | Trop. | Table 2 | 15 | 3.1 | NH, this study |
| EMEP | Trop. | Table 2 | 16 | 2.4 | globe, this study |
| | | | | | |
| **Winter** | | | | | |
| Den93 | Trop. | 0.1 | 75[a] | 20 | NH, Nov-Apr. |
| Den93 | Trop. | 0.01 | 66[a] | 12 | NH, Nov-Apr. |
| Tie01 | Trop. | 0.1[b] | 73 | 11 | 45°N, Dec. |
| Tie01 | Trop. | 0.1[b] | 3 | 3 | Equator, Dec. |
| Tie03 | Trop. | 0.1[b] | 38 | 6 | globe, Dec. |
| Tie03 | Trop. | 0.1[b] | 47 | 7 | NH, Dec. |
| HAMMOZ | Trop. | Table 2 | 9.1 | 2.0 | globe, Dec-Feb., this study |
| HAMMOZ | Trop. | Table 2 | 24. | 3.8 | NH, Dec-Feb., this study |
| HAMMOZ | surface | Table 2 | 16. | 8 | globe, Dec-Feb., this study |
| EMEP | surface | Table 2 | 5.1 | 4.9 | globe, Dec-Feb., this study |
| | | | | | |
| **Summer** | | | | | |
| Den93 | Trop. | 0.1 | 45[a] | 13 | NH, May-Oct. |
| Den93 | Trop. | 0.01 | 30[a] | 5 | NH, May-Oct. |
| Tie01 | Trop. | 0.1[b] | 7 | 7 | 45°N, June |
| Tie01 | Trop. | 0.1[b] | 2 | 2 | Equator, June |
| Tie03 | Trop. | 0.1[b] | 6 | 4 | globe, Dec. |
| Tie03 | Trop. | 0.1[b] | 7 | ∼5.5 | NH, Dec. |
| EMEP | surface | Table 2 | 2.3 | 3.0 | globe, Jun-Aug , this study |
| HAMMOZ | surface | Table 2 | 0.4 | 2 | NH, Jun-Aug, this study |
| HAMMOZ | Trop. | Table 2 | 2.4 | 1.3 | NH, Jun-Aug, this study |

Refs: Den93: Dentener and Crutzen (1993) ,ME10: Macintyre and Evans (2010) ,Tie01: Tie et al. (2001) ;

Notes: a.Dentener and Crutzen (1993) reported changes in NO$X$=NO+NO2+NO3+2N2O5+HNO4, not NOx; b. Tie et al. (2001) used surface area of sulfate aerosols only; Trop. denotes full model domain, e.g. 0–100 hPa for EMEP, 0–4 hPa for Tie et al. (2001); NH denotes northen hemisphere. Data extracted from figures by eye indicated with ∼. approximate.

with 100 times the area of the $1\times1°$ grid used in EMEP or almost 30 times that of ECHAM-HAMMOZ's $1.85\times1.85°$ grid. This alone will lead to different regimes of ozone productivity. It can also be noted that global CTMs (including changes due to emissions and chemical mechanisms) have improved over the years, so recent models should be expected to have different sensitivities to earlier studies (Wu et al., 2007). Emissions have also changed enormously over this period, especially in Asia (Granier et al., 2011); again with implications for the atmospheric oxidation capacity.

There are many other aspects of heterogeneous chemistry which are potentially important, but extremely complex and beyond the scope and abilities of our models. This includes for example the strong interactions of $NO_2$ with aerosol water and sulphate formation seen in wintertime haze events in Beijing (Cheng et al., 2016). However, Cheng et al. (2016) were concerned with extreme aerosol pollution events with concentrations exceeding 100 $\mu g$ m$^{-3}$. These cannot be modeled at present in global scale models because of the dilution effect of the coarse grid resolution, and such extreme pollution events are likely to only have a local importance. In any case, it is not certain that the mechanism suggested by Cheng et al. (2016), is sufficient to explain some other extreme smog events (e.g. Guo et al., 2017).

Another important aspect for $N_2O_5$ heterogeneous chemistry is the formation of $ClNO_2$. In this case, $N_2O_5$ reacts with particulate chlorine to form gas-phase $ClNO_2$, which can photolyse and recycle $NO_2$ (Wang et al., 2016) and alter $NO_y$ composition (Sarwar et al., 2012). Especially in the planetary boundary layer of southern China, high mixing ratios of $ClNO_2$ ($> 400$ pptv) and $N_2O_5$ ($> 1$ ppbv) have been observed (Wang et al., 2016). The formation of $ClNO_2$ lowers the impact of $N_2O_5$ reaction on ozone, because it recycles $NO_2$ and was observed to enhance the ozone peak in southern China up to 16% (Wang et al., 2016). Unfortunately, our models (and indeed most global models) lack chlorine chemistry and treatment of chlorine in the aerosol thermodynamics, so cannot tackle these issues.

Heterogeneous reactions on cloud surfaces, which can be important especially for $HO_2$ uptake depending of the presence of transition metal ions, were also excluded from our study. However, Dentener and Crutzen (1993) included the reaction of $N_2O_5$ on cloud droplets, but just found minor changes in $NO_x$ and $O_3$. Jacob (2000) argue that for $O_3$, $HO_x$ and $NO_x$, life times are not significantly reduced in clouds and current knowledge is insufficient to include cloud chemistry in $O_3$ models. In fact, most global model studies exclude the heterogeneous reactions of nitrogen species on clouds. Therefore, further development of CTM cloud-chemical systems will be needed before this question can be properly addressed.

In summary, our study finds a lower but still important impact of $N_2O_5$ hydrolysis on ozone and nitrogen oxides compared to previous model studies. However, earlier studies used rather high $\gamma$ values for $N_2O_5$, and neglected the other heterogeneous reactions. Further, chemical transport models have developed in many ways over the last 20-30 years, and indeed emissions across the globe have dramatically changed over this time period. In this paper we have illustrated that ECHAM-HAMMOZ and EMEP, two up-to-date models systems, are rather consistent in the importance of $N_2O_5$ reactions, and that such reactions seem to be the most important among the six reactions we tested. Although one can never know if models produce good results for the right reasons, we have shown that both ECHAM-HAMMOZ and EMEP can reproduce even daily ozone variations remarkably well at sites across the globe (one can contrast results for Mace Head between Fig. 7 and the wide range of data from earlier models presented in Wild et al. 2012). We have also demonstrated that both models do a fair job of reproducing

surface area density, so we believe our new estimates provide a valuable revision of calculations concerning the impact of heterogeneous reactions in such CTMs.

## 6 Conclusions

Two global transport models were used to investigate the implications of six heterogeneous (gas-aerosol uptake) reactions on ground-level ozone concentrations. Both models were harmonized to use similar parameterizations for most of these reactions, enabling us to compare the impacts of $N_2O_5$, $NO_3$, $NO_2$, $O_3$, $HNO_3$, and $HO_2$ on ozone mixing ratios. Each reaction was evaluated systematically comparing the reference run to sensitivity simulations excluding one reaction at a time. Since heterogeneous reactions take place at the aerosol surface area, the modeled surface area density $S_a$ of both models was compared to a satellite product retrieving the surface area. This comparison shows a good agreement in global pattern and especially the capability of both models to capture the extreme aerosol loadings in East Asia.

The analysis of the sensitivity runs confirms that the globally most important heterogeneous reaction is the one of $N_2O_5$. This impact was expected from previous studies, with the surface reactions of $N_2O_5$ having an impact on ozone mixing ratios through removal of reactive $NO_x$ species. This result is loosely consistent with results from earlier studies (e.g. Dentener and Crutzen, 1993; Tie et al., 2001, 2003; Alexander et al., 2009; Macintyre and Evans, 2010), although here the magnitude of changes induced by $N_2O_5$ reaction is at the low end of estimates, which seems to fit a trend whereby the more recent the study the lower the impacts of these reactions. Some other heterogeneous reactions (especially the ones of $NO_2$, $HO_2$ and $HNO_3$) gain some significance in highly polluted areas where aerosol surface areas are high, but the two models show quite different response in their response to these other gas-aerosol reactions. The EMEP model actually shows rather small impacts of these reactions, except in East and South Asia where some impacts can approach 10-20% of that of $N_2O_5$. ECHAM-HAMMOZ, on the other hand, shows quite marked responses to especially the $HNO_3$ reactions. The reasons for this are related to differences in nitrate chemistry and surface area assumptions in the models, and to the differing spatial resolutions. It may well be that ECHAM-HAMMOZ overestimates the impact of $HNO_3$ due to missing nitrate aerosol formation and EMEP underestimates the impact, due to the use of only coarse sea salt and dust aerosol for the $HNO_3$ and $HO_2$ reactions.

The reactions of $O_3$ on dust and $NO_3$ on aerosols were found to have only minor effects on ozone in comparison to the other reactions in both models. In terms of global spatial impact, all reactions related to nitrogen species alter atmospheric chemistry downwind of source areas to some extent, with changes being much larger in the polluted northern hemisphere than in the southern hemisphere.

Evaluation of the models with northern hemispheric ozone surface observations from the GAW/TOAR networks yields a better agreement of the models with observations in terms of daily maximum concentrations, variability and temporal correlations at most sites when the heterogeneous reactions are incorporated. The impacts of the $N_2O_5$ reactions show strong seasonal variations, with biggest impacts in spring time when photochemical reactions are active and $N_2O_5$ levels still high.

Due to lack of direct observations substantial uncertainties remain regarding the impact of heterogeneous reactions on tropospheric reactive gases. It should be noted, that neither model had an implementation of the particle-liquid-water/nitrate/chloride

effects suggested by Bertram and Thornton (2009) and tested by e.g. Lowe et al. (2015). Further, neither model includes halogen chemistry, which is also known to impact $O_3$ in polluted regions (e.g. Sarwar et al., 2014; Li et al., 2016). The large impact of $N_2O_5$ seen in our work might be somewhat overestimated compared to that we would obtain if the chemistry of $ClNO_2$ (which would recycle $NO_x$) and other halogens could be included. Such improvements should result in better particle phase

5   chemistry, and will be the subject of future work.

*Acknowledgements.* The authors wish to thank the Jülich Supercomputing Centre (2016) for providing the computing resources for the ECHAM-HAMMOZ simulations. The EMEP work was funded by the EU FP7 projects ECLAIRE (Project number 282910), and EMEP under UNECE, with computer time supported by the Research Council of Norway (Programme for Supercomputing). The project was also supported by the Swedish Climate Modelling Research Project MERGE. GAW surface ozone observation data were retrieved from the

10   World Data Center for Greenhouse Gases in Tokyo, Japan. We acknowledge the substantial efforts of all data providers for making these measurements.

*Competing interests.* No competing interests are present

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
