# Peer review of "Ozone Impacts of Gas-Aerosol Uptake in Global Chemistry Transport Models"

_Atmospheric Chemistry and Physics, 2017_

## Referee Comment (RC1) · Anonymous Referee #1 · 17 Jul 2017

The manuscript "Ozone Impacts of Gas-Aerosol Uptake in Global Chemistry Transport Models" by Stadtler et al. deals with the importance of heterogeneous chemical uptake of several inorganic compounds for the tropospheric chemical budget of ozone. To investigate the relevance of the uptake, parameterised reaction rates producing a loss term to the budget of $N_2O_5$, $NO_3$, $NO_2$, $O_3$, $HNO_3$ and $HO_2$ are implemented in two chemistry models, i.e. the EMEP chemical transport model and the ECHAM-HAMMOZ CCM. The analysis of the results is based on sensitivity tests, each neglecting one of the reactions. The dominating heterogeneous reaction for the tropospheric $O_3$ burden is the (well known) heterogeneous uptake of $N_2O_5$, whereas the other reactions show a minor relevance. Given the large temporal variability of the ozone measurements at the given stations, even the heterogeneous $N_2O_5$ uptake is hardly significant, whereas

for the other reactions a statistical significance cannot be stated. For some reactions the two models even give results of the opposite sign.

The paper is well written and the study is reasonable and should be published in ACP after addressing the comments below. However, as the main finding is the importance of the $N_2O_5$ uptake, which is well known since more than 20 years, the new findings from this manuscript are relatively limited.

Major comments:

1) The neglect of direct nitrate formation in HAMMOZ might not only lead to a shift between chloride release and nitrate formation, but might affect also the size distribution and therefore the surface area of the aerosol. To which degree does the SAD change in the EMEP simulations in case of the neglect of $N_2O_5$ or $HNO_3$ uptake? Can you quantify the total loss of these compounds in the EMEP simulations versus the HAMMOZ runs?

2) To which degree do you think that reactive nitrogen chemistry as in '"Reactive nitrogen chemistry in aerosol water as a source of sulfate during haze events in China": Yafang Cheng, Guangjie Zheng, Chao Wei, Qing Mu, Bo Zheng, Zhibin Wang, Meng Gao, Qiang Zhang, Kebin He, Gregory Carmichael, Ulrich Pöschl, Hang Su Science Advances, 21 Dec. 2016; DOI: 10.1126/sciadv.1601530' is relevant on the global scale in your simulations?

3) The interaction of the individual uptake reactions might lead to a masking of the direct effects of the reactions by chemical interactions. Why did you not choose to analyse individual reactions, but always the sum of reactions neglecting only one of them. Would a method such as a Factor Separation Method (see Stein, U. and P. Alpert, 1993: Factor Separation in Numerical Simulations. J. Atmos. Sci., 50, 2107–2115, https://doi.org/10.1175/1520-0469(1993)050<2107:FSINS>2.0.CO;2) not be more suitable to address your scientific question (given the extra computation time is available).

4) To which degree might your results depend on the selected year. Would you expect substantial inter-annual variability based on the variability in both the aerosol surface and the constituents?

5) How sensitive are your results to the selected $\gamma$ values for the individual compounds? Are the values important or is the uptake anyhow mostly determined by the available aerosol surface and the availability of the constituents?

6) Why did you only consider $O_3$ uptake on dust and not on liquid aerosol particles, where it can contribute to "in-aerosol" sulphate formation by oxidation of dissolved $SO_2$. Even though the solubility of $O_3$ is quite low, the effective uptake is determined by the reaction rate.

7) What is the tropospheric $CH_4$ lifetime in your simulations? Due to $HO_2$ uptake, the oxidation capacity might be reduced, but due to $HONO$ formation and subsequent photolysis otherwise affected. Is this similar in both models? Is there a substantial impact on the oxidation capacity which is not that obvious in the well buffered compound $O_3$?

Technical comments:

1) Fig. 6 to 8: The grey shaded area is very difficult to see. Please think about a better visualisation of the observations.

2) page 7, last line: citation is missing/wrong (ibid)

3) page 13, line 7 (wrong line number): citation is missing/wrong (ibid) "transport model, GEOS-Chem (ibid),....."

---

## Referee Comment (RC2) · Anonymous Referee #2 · 16 Sep 2017

**General Comments:**

This paper presents two global model simulations (EMEP, ECHAM-HAMMOZ) that assess the influence of six heterogeneous reactions on global atmospheric mixing ratios of reactive nitrogen species and ozone. The six reactions (given in Table 1) have been investigated in previous model studies, notably Jacob, Atmos. Environ. 2000, on which this submission appears to be largely based. The influence of heterogeneous chemistry is known to be important in global chemical transport models, but it is also generally difficult to parameterize for a number of reasons. These include the difficultly of accurately simulating aerosol surface areas available for heterogeneous reactions and the large uncertainty in some uptake coefficients, especially N2O5, which has the largest effects in the analysis from this paper. For these reasons, further investigations

of the details of the heterogeneous reactions in global models are generally valuable contributions to the literature.

While the above is a good justification for the present work, it is somewhat difficult to see that this paper advances the subject much beyond what has been presented in previous papers. This deficiency in presentation could likely be addressed, but the authors would do well to review how their results compare to previous model simulations that have investigated this set of reactions previously, as well as explicitly stating how their treatment differs and why their model arrives at different results or corroborates previous analysis. Such a comparison is and critical model evaluation is absent from the discussion section. The paper would be much stronger if it were included.

There is no discussion of the effects of clouds, which have large surface areas for heterogeneous reactions. Are all simulations showing effects of reactions on aerosols but not in clouds? This should be clarified, together with some estimate of the relative effects of both cloud droplets and aerososls if they are both operative in the models.

The paper identifies N2O5 uptake as the most important heterogeneous reaction of the six, but it does not include the production of CINO2 from N2O5 uptake. The authors state this deficiency clearly, but at that same time it is a missed opportunity since it would be one aspect where these model simulations could clearly take advantage of recent advances in field and laboratory work. No real explanation is given as to the "technical details" that prevent the inclusion of this reaction, but the omission should be better justified. Even a crude estimate of this reaction would be helpful to this analysis.

Although the paper does not appear to represent a significant advance (unless the authors provide some further comparisons and details), it does not appear to be incorrect in any obvious way. There are some issues with presentation, detailed below, but these issues do not appear to be serious. With some attention to the comments above and the more specific comments below, it should be suitable for publication in ACP.

Specific Comments:

Page 2, line 1: Solomon et al., Nature 1986 is a better reference (suggestion only) Âă Page 2, line 3: Ravishankara 1997 is a better reference here (again, suggestion only). Ravishankara, A. R. (1997), Heterogeneous and multiphase chemistry in the troposphere, Science, 276, 1058-1065. Âă Page 2, line 22: Not clear what is meant by "technical limitations" here that precludes the inclusion of CINO2.Âă Âă Page 4, first paragraph: The term "Sa" is used to refer to aerosol surface area, but later in the paragraph "S" is used as total surface area. Is there a distinction between S and Sa, or is this just typographic.

Page 4, line 17: correct grammar in "make use"

Page 5, lines 8-13: Does the lack of nitrate aerosol formation artificially reduce aerosol surface area available for heterogeneous chemistry?

Page 6, line 15: The value of 5e-1 (0.5) must be misquoted as the N2O5 reaction probability is not this large.

Page 7, lines 14-16: The authors make a good point regarding the reliability of the parameterization, especially in light of the absence of atmospheric determinations of gamma values as high as those shown in figure 1. Have the authors made any assessment of the effect of reducing the RH dependence of the parameterization for sulfate?

Page 7, section 3.2: The literature cited for the gamma value of NO3 is dated. More recent work by Gross et al. shows substantial reactivity on organics, for example. There is somewhat less active research in this area than for uptake coefficients for N2O5, so there is no developed parameterization. It is not realistic for the authors to undertake such a review in the context of this paper, but some reference to the more recent studies together with a statement that NO3 uptake may be larger if organic aerosol is considered is needed in this section.

Gross, S., and A. K. Bertram (2008), Reactive Uptake of NO3, N2O5, NO2, HNO3, and O3 on Three Types of Polycyclic Aromatic Hydrocarbon Surfaces, The Journal

of Physical Chemistry A, 112(14), 3104-3113, 10.1021/jp7107544. Gross, S., and A. K. Bertram (2009), Products and kinetics of the reaction of an alkane monolayer and a terminal alkene monolayer with NO3 radicals, J. Geophys. Res., 114, D02307, doi:10.1029/2008JD010987. Gross, S., R. Q. Iannone, S. Xiao, and A. K. Bertram (2009), Reactive uptake studies of NO3 and N2O5 on alkenoic acid, alkanoate and polyalcohol substrates to probe nighttime aerosol chemistry, Phys. Chem. Chem. Phys., 11, 7792-7803.

Page 7, section 3.3: Again, the authors are justified in the use of the simple uptake coefficient for NO2 based on what is currently available in the literature, but the system is at least as complex as that for N2O5. Some model studies have assumed effectively very large uptake coefficients for NO2 or at the very least rapid conversion of NO2 to HONO. This body of literature should be represented here via referencing. One example of a recent modeling study: Elshorbany, Y. F., P. J. Crutzen, B. Steil, A. Pozzer, H. Tost, and J. Lelieveld (2014), Global and regional impacts of HONO on the chemical composition of clouds and aerosols, Atmos. Chem. Phys., 14(3), 1167-1184, 10.5194/acp-14-1167-2014.

Page 13, section 5.1 and Figure 2: What altitude range is shown in Figure 2? Is this for some distance above the surface, boundary layer only, column average, etc? Second, is the displayed quantity a dry aerosol surface area or does it include water? If the latter (presumably), to what extent are the regional variations due to RH and to what extent to dry aerosol mass?

Table 4: Caption states that reference runs values are given in "total mixing ratios." This term is not clear. Do the authors mean average? Once again, over what altitude range do these values apply? Was this information given elsewhere? If so, it should be repeated here as it is not clear when reading the table or figure 2.

The caption also appears to be logically in error: "Since the sensitivity runs were subtracted from the reference run, positive values mean higher mixing ratios in the reference run than in the sensitivity runs and vice versa". By this logic, a higher value in the reference run would lead to a negative displayed value, consistent with what is shown in the table (e.g., removing O3 uptake should increase O3 in the sensitivity run, leading to sensitivity > reference, or reference – sensitivity < 0, as shown for "no O3").

It is also notable in this table that the change in N2O5 are larger than the "total" or average. How can this be?

The tables are somewhat difficult to interpret since they are in absolute units. Relative changes (e.g., -10%, +20%) would be easier to understand. Relative changes should at least be given in addition to the absolute changes, and could be substituted for them easily since the absolute value is given in each case for the reference run.

Page 17, line 25: The meaning if the sentence is not clear. "NO3 rapidly photolyses, and resulting NO2 likewise, so has a high ozone-formation potential."

Figure 3,4: Again, please specify the altitude range in the captions and text.

Page 18, lines 4-11: How do the reductions in O3 and NOx compare with those determined from other model studies, e.g., Dentener and Crutzen (1993), Tie et al. (2001, 2003), Alexander et al. (2009), Macintyre and Evans (2010) etc. Critical comparisons of these results to these and other literature studies are missing, but extremely important to place the current work in context and understand what advances have been made in this model analysis.

As with Tables 4 and 5, these figures would be more easily interpreted in relative units (% change in O3) rather than absolute units (ppbv) as shown.

Page 18, line 25: Suggest a change in the phrase "therefore the sun is less favorable" to something more like "therefore photochemistry is inactive".

Page 18, line 28: The statement is that "less ozone production reduction occurs" (awkward phrasing). Is this a statement about absolute or relative ozone production? The latter would be more relevant, since it is already understood that ozone photochemistry

is weaker in winter.

Page 22, line 9: The comma should be after "ground stations". Having it before ground stations changes the meaning of the sentence in a way that the authors probably do not intend.

---

## Author Comment (AC1) · 30 Nov 2017

**1 Anonymous Referee #1**

**1.1 General comments**

The manuscript "Ozone Impacts of Gas-Aerosol Uptake in Global Chemistry Transport Models" by Stadtler et al. deals with the importance of heterogeneous chemical uptake of several inorganic compounds for the tropospheric chemical budget of ozone. To investigate the relevance of the uptake, parameterised reaction rates producing a loss term to the budget of $N_2O_5$, $NO_3$, $NO_2$, $O_3$, $HNO_3$ and $HO_2$ are implemented in two chemistry models, i.e. the EMEP chemical transport model and the ECHAM-HAMMOZ CCM. The analysis of the results is based on sensitivity tests, each neglecting one of the reactions. The dominating heterogeneous reaction for the tropospheric $O_3$ burden is the (well known) heterogeneous uptake of $N_2O_5$, whereas the other reactions show a minor relevance. Given the large temporal variability of the ozone measurements at the given stations, even the heterogeneous $N_2O_5$ uptake is hardly significant, whereas for the other reactions a statistical significance cannot be stated. For some reactions the two models even give results of the opposite sign. The paper is well written and the study is reasonable and should be published in ACP after addressing the comments below. However, as the main finding is the importance of the $N_2O_5$ uptake, which is well known since more than 20 years, the new findings from this manuscript are relatively limited.

    Reply: We thank the referee for the positive and constructive comments. Concerning the last point, we agree that the importance of $N_2O_5$ has been established for many years, but we believe that our paper is novel in using very up-to-date chemical transport models (CTMs) which we show can reproduce well daily variations at sites around the world, and with a demonstration of a fair ability to capture aerosol surface area compared to satellite data. In addition, we also illustrate in detail how seasonal patterns are affected by this reaction. We also believe we are the first to systematically compare the impacts of the different reactions in a harmonized way across two model systems. We have now added also a comparison to previous models studies and also new sensitivity tests on the impact of different assumptions for $\gamma$. Details can be found in the new Discussion Section 5.4.

    Before giving the main replies below, we should point out that for the revised manuscript we re-ran all results with an updated version of the EMEP model. This was done following some

bug-fixes in the rv4.15 version originally used, including in the deposition of $N_2O_5$ and radiation schemes. These changes have not affected basic model performance very much, but especially the deposition change affects $N_2O_5$ levels, and the impact of the noN2O5 scenarios. Indeed, the impacts in the EMEP system now resemble much more those of ECHAM-HAMMOZ. We have added a small Appendix to explain these changes, and modified the manuscript to reflect the updated results.

**1.2   Major comments**

1) The neglect of direct nitrate formation in HAMMOZ might not only lead to a shift between chloride release and nitrate formation, but might affect also the size distribution and therefore the surface area of the aerosol. To which degree does the SAD change in the EMEP simulations in case of the neglect of $N_2O_5$ or $HNO_3$ uptake? Can you quantify the total loss of these compounds in the EMEP simulations versus the HAMMOZ runs?

   Reply: As shown in Figure 5 SAD hardly changes in EMEP simulations with the neglect of $N_2O_5$ reactions, and a similar pattern is seen for $HNO_3$ reactions (not shown).
   Unfortunately the models do not track the various loss terms so we cannot compare the total losses in EMEP versus ECHAM-HAMMOZ. We agree that this would be very valuable information, but it would require re-writing the model code and re-running all simulations.

   2) To which degree do you think that reactive nitrogen chemistry as in Reactive nitrogen chemistry in aerosol water as a source of sulfate during haze events in China: Yafang Cheng, Guangjie Zheng, Chao Wei, Qing Mu, Bo Zheng, Zhibin Wang, Meng Gao, Qiang Zhang, Kebin He, Gregory Carmichael, Ulrich Pschl, Hang Su Science Advances, 21 Dec. 2016; DOI: 10.1126/sciadv.1601530 is relevant on the global scale in your simulations?

   Reply:   It is actually very hard to answer this question. The strong uptake and impact of $NO_2$ in aerosol water systems as explored by Cheng et al. is not included in either of our models. However, Cheng et al. were concerned with extreme aerosol pollution events with concentrations exceeding ($100 \mu$ g m$^{-3}$. These cannot be modeled at present in global scale models because of the dilution effect of the coarse grid resolution, and such extreme pollution events are likely to only have a local importance. Finally, it is not certain that the mechanism suggested by Cheng et al., is sufficient to explain some other extreme smog events (Guo et al. 2017 doi:10.1038/s41598-017-11704-0). As well as considerable scientific uncertainties in gas-aerosol interactions in such extreme conditions, proper implementation of these type of effects would depend on very good estimates of all emissions (including base-cations), air-liquid interactions, activity coefficients, pH, etc.; factors which are beyond the capabilities of current global atmospheric chemistry models.
   We have added some extra words on these problems in a new Discussion Section, 5.4.

3) The interaction of the individual uptake reactions might lead to a masking of the direct effects of the reactions by chemical interactions. Why did you not choose to analyse individual reactions, but always the sum of reactions neglecting only one of them. Would a method such as a Factor Separation Method (see Stein, U. and P. Alpert, 1993: Factor Separation in Numerical Simulations. J. Atmos. Sci., 50, 21072115, https://doi.org/10.1175/1520-0469(1993)050¡2107:FSINS¿2.0.CO;2) not be more suitable to address your scientific question (given the extra computation time is available).

Reply: Our procedure is to start with the reference run which provides the best estimate of the photochemical system we can make with our models, and to then eliminate each reaction one by one. Thus, each change is a simple perturbation due to that individual reaction, which we believe gives both a fair and simple comparison across the reactions. For $n$ factors, the Factor Separation Method of Stein and Alpert requires $2^n$ simulations. Thus, our 6 reactions would have required 64 simulations, which would have been prohibitive for ECHAM-HAMMOZ at least. Further, the changes we calculate are generally very small for most reactions, so a linear (perturbation) analysis should be both clearer and accurate.

4) To which degree might your results depend on the selected year. Would you expect substantial inter-annual variability based on the variability in both the aerosol surface and the constituents?

Reply: New tests have been conducted with the EMEP model for 2011, and in Tables 1 and 2 below we compare the percentage changes of this year to those of 2012. It can be seen that the results are remarkably similar, across all compounds and reaction tests. This suggests that the strong day-to-day variability in atmospheric and pollution conditions, and impacts of these reactions, average out to a large extent and annual results are rather robust.

5) How sensitive are your results to the selected values for the individual compounds? Are the values important or is the uptake anyhow mostly determined by the available aerosol surface and the availability of the constituents?

Reply: This is a complicated question, since these reactions also change the composition of $NO_y$ in the atmosphere, the lifetime of $NO_2$ and hence the photooxidation processes leading to $O_3$. In order to address this, we have run the EMEP model with four new configurations:

1. $\gamma = 0.01$ for $N_2O_5$, a value lower than our typical values, and at the low end of estimates (manuscript Sect. 3.1)

2. $\gamma = 0.1$ for $N_2O_5$, equivalent to values used by e.g. Dentener and Crutzen (1993), which is substantially higher than our values

3. $\gamma = 1.0 \times 10^{-3}$ for $NO_2$, at the top end of estimates (Sect. 3.3)

Table 1: Impacts of gas-aerosol reactions on regional average mixing ratios of $O_3$ and key NOy compounds: EMEP model, year 2011.

| Region | Run | Unit | $O_3$ | $NO_x$ | $NO_y$ | $HNO_3$ | PAN | $N_2O_5$ | $NO_3$ |
|--------|-----|------|-------|--------|--------|---------|-----|----------|--------|
| NA | Base | Conc*: | 39.85 | 0.80 | 1.79 | 0.20 | 0.54 | 4.50 | 4.09 |
| NA | noN2O5pm | %: | 6 | 10 | 4 | -11 | 9 | 202 | 70 |
| NA | noHO2pm | %: | 0 | -1 | 0 | 1 | 1 | -2 | -11 |
| NA | noHNO3pm | %: | 0 | 0 | -3 | 23 | 0 | 0 | 0 |
| NA | noNO2pm | %: | 0 | 2 | 1 | 0 | 0 | 4 | 2 |
| NA | noNO3pm | %: | 0 | 0 | 0 | 0 | 0 | 1 | 4 |
| NA | noO3pm | %: | 0 | 0 | 0 | 0 | 0 | 0 | 0 |
| | | | | | | | | | |
| EUR | Base | Conc*: | 40.15 | 0.99 | 2.40 | 0.25 | 0.50 | 6.98 | 5.90 |
| EUR | noN2O5pm | %: | 7 | 17 | 3 | -16 | 11 | 316 | 86 |
| EUR | noHO2pm | %: | 1 | -3 | 0 | 1 | 4 | -4 | -14 |
| EUR | noHNO3pm | %: | 1 | 0 | -6 | 59 | 0 | 0 | 3 |
| EUR | noNO2pm | %: | 0 | 5 | 1 | -1 | -1 | 6 | 4 |
| EUR | noNO3pm | %: | 0 | 0 | 0 | 0 | 0 | 2 | 10 |
| EUR | noO3pm | %: | 0 | 0 | 0 | 0 | 0 | 0 | 2 |
| | | | | | | | | | |
| EA | Base | Conc*: | 42.65 | 2.17 | 4.50 | 0.54 | 0.79 | 10.88 | 4.62 |
| EA | noN2O5pm | %: | 8 | 15 | 5 | -19 | 13 | 316 | 127 |
| EA | noHO2pm | %: | 2 | -4 | 0 | 2 | 7 | 0 | -8 |
| EA | noHNO3pm | %: | 0 | 0 | -2 | 16 | 0 | 0 | 0 |
| EA | noNO2pm | %: | -1 | 29 | 9 | -11 | -7 | 12 | 3 |
| EA | noNO3pm | %: | 0 | 0 | 0 | 0 | 0 | 1 | 3 |
| EA | noO3pm | %: | 0 | 0 | 0 | 0 | 0 | 0 | 0 |
| | | | | | | | | | |
| SA | Base | Conc*: | 47.12 | 1.09 | 2.83 | 0.40 | 0.29 | 8.78 | 11.27 |
| SA | noN2O5pm | %: | 7 | 12 | 1 | -4 | 15 | 175 | 73 |
| SA | noHO2pm | %: | 1 | -3 | 0 | 1 | 5 | -5 | -12 |
| SA | noHNO3pm | %: | 1 | 0 | -8 | 63 | 0 | 1 | 4 |
| SA | noNO2pm | %: | 1 | 4 | 1 | 0 | 1 | 9 | 5 |
| SA | noNO3pm | %: | 1 | 0 | 0 | 0 | 1 | 5 | 10 |
| SA | noO3pm | %: | 0 | 0 | 0 | 0 | 0 | 0 | 0 |

Notes: Base-case concentrations from the surface-level of the model are given in ppt for $NO_3$ and $N_2O_5$, otherwise ppb (Conc* flags this difference in units). Results for the sensitivity tests are given as (test-base)/base in %. The first column refers to the region over which the annual mean is spatially averaged, and the second column refers to the corresponding run. Regions are defined as follows: NA (15° N55° N; 60° W125° W), EU (25 ° N65° N; 10° W-50° E), EA (15° N50° N; 95° E160° E), and SA (5° N35° N; 50° E95° E).

Table 2: Impacts of gas-aerosol reactions on regional average mixing ratios of $O_3$ and key NOy compounds: EMEP model, year 2012.

| Region | Run | Unit | $O_3$ | $NO_x$ | $NO_y$ | $HNO_3$ | PAN | $N_2O_5$ | $NO_3$ |
|--------|-----|------|-------|--------|--------|---------|-----|----------|--------|
| NA | Base | Conc*: | 40.33 | 0.82 | 1.81 | 0.21 | 0.55 | 5.08 | 4.54 |
| NA | noN2O5pm | %: | 5 | 9 | 4 | -10 | 8 | 160 | 59 |
| NA | noHO2pm | %: | 0 | -1 | 0 | 1 | 1 | -2 | -10 |
| NA | noHNO3pm | %: | 0 | 0 | -2 | 18 | 0 | 0 | 0 |
| NA | noNO2pm | %: | 0 | 2 | 1 | -1 | 0 | 4 | 2 |
| NA | noNO3pm | %: | 0 | 0 | 0 | 0 | 0 | 1 | 3 |
| NA | noO3pm | %: | 0 | 0 | 0 | 0 | 0 | 0 | 0 |
| | | | | | | | | | |
| EUR | Base | Conc*: | 40.89 | 1.01 | 2.43 | 0.25 | 0.54 | 7.73 | 6.48 |
| EUR | noN2O5pm | %: | 7 | 16 | 3 | -16 | 10 | 280 | 72 |
| EUR | noHO2pm | %: | 1 | -3 | 0 | 1 | 4 | -4 | -14 |
| EUR | noHNO3pm | %: | 1 | 0 | -6 | 58 | 0 | 0 | 3 |
| EUR | noNO2pm | %: | 0 | 5 | 1 | -1 | -1 | 6 | 4 |
| EUR | noNO3pm | %: | 0 | 0 | 0 | 0 | 0 | 2 | 10 |
| EUR | noO3pm | %: | 0 | 0 | 0 | 0 | 0 | 0 | 2 |
| | | | | | | | | | |
| EA | Base | Conc*: | 43.96 | 2.23 | 4.63 | 0.54 | 0.89 | 12.59 | 5.52 |
| EA | noN2O5pm | %: | 8 | 14 | 4 | -19 | 13 | 278 | 106 |
| EA | noHO2pm | %: | 2 | -4 | 0 | 2 | 7 | 0 | -7 |
| EA | noHNO3pm | %: | 0 | 0 | -2 | 13 | 0 | 0 | 0 |
| EA | noNO2pm | %: | -1 | 30 | 9 | -11 | -8 | 13 | 4 |
| EA | noNO3pm | %: | 0 | 0 | 0 | 0 | 0 | 1 | 3 |
| EA | noO3pm | %: | 0 | 0 | 0 | 0 | 0 | 0 | 0 |
| | | | | | | | | | |
| SA | Base | Conc*: | 47.33 | 1.12 | 2.90 | 0.42 | 0.33 | 10.37 | 12.04 |
| SA | noN2O5pm | %: | 6 | 11 | 1 | -4 | 15 | 139 | 63 |
| SA | noHO2pm | %: | 1 | -3 | 0 | 1 | 5 | -5 | -12 |
| SA | noHNO3pm | %: | 1 | 0 | -8 | 61 | 0 | 1 | 4 |
| SA | noNO2pm | %: | 1 | 4 | 1 | 0 | 1 | 10 | 5 |
| SA | noNO3pm | %: | 1 | 0 | 0 | 0 | 1 | 5 | 11 |
| SA | noO3pm | %: | 0 | 0 | 0 | 0 | 0 | 0 | 0 |

Notes: Base-case concentrations from the surface-level of the model are given in ppt for $NO_3$ and $N_2O_5$, otherwise ppb (Conc* flags this difference in units). Results for the sensitivity tests are given as (test-base)/base in %. The first column refers to the region over which the annual mean is spatially averaged, and the second column refers to the corresponding run. Regions are defined as follows: NA (15° N55° N; 60° W125° W), EU (25 ° N65° N; 10° W-50° E), EA (15° N50° N; 95° E160° E), and SA (5° N35° N; 50° E95° E).

4. $\gamma = 0.0$ for $NO_2$, since the lowest estimates are extremely low.

The model has been run for new base-cases, and for the noN2O5, noHNO3 and (except for test 4) noNO2 cases. Results for the regional averages (equivalent to manuscript Tables 4-5) are shown in Tables 3-4 below. Considering the $N_2O_5$ tests first, then the changes in ozone over for example North America range from 3% ($\gamma = 0.01$) to 8% ($\gamma = 0.1$), compared to our original estimate of 5% (Table 4). Changes for $NO_x$ follow a similar pattern (e.g. 6-13% for NA, versus original 9%), but changes for $N_2O_5$ itself are much more significant (80% versus 354%, compared to the original 160%).

Considering the $\gamma$ tests for $NO_2$, then again the test results for the noN2O5 tests generally span those of the original runs, e.g. changes of 4-6% for ozone in North America versus 5% in the original run, or 113-170% for $N_2O_5$ versus 160% for the original case. Test (3), with the high $\gamma = 1.0\times10^{-3}$ for $NO_2$ does have significant impacts on the $NO_x$ levels though, from e.g. 2% in the original run to 16% in test (3) for NA, or from 30% to 109% in East Asia. In these runs the impacts of noNO2 on ozone become comparable to those of noN2O5, and in South Asia the ozone changes from noNO2 actually exceed those from noN2O5.

Test (4), using zero $\gamma$ actually gives results which are very similar to our default $\gamma = 1.0\times10^{-4}$, suggesting that this reaction only becomes important if higher values than $\times10^{-4}$ can be justified.

Thus, we find that the exact changes in ozone and N-compounds do depend on the assumed $\gamma$ values, but the relative importance of the different heterogeneous reactions generally remains. The $N_2O_5$ reactions are in nearly all cases the most important driver of ozone changes, but the use of a very high values for $\gamma$ for $NO_2$ changes the picture somewhat. We can note though that use of the high 0.001 values for $\gamma(NO_2)$ leads to quite significant reductions in annual $NO_2$ concentrations (not shown), resulting in degraded performance of the EMEP model compared to measurements, at least across the EMEP observational network in Europe (Tørseth et al. 2012).

6) Why did you only consider $O_3$ uptake on dust and not on liquid aerosol particles, where it can contribute to "in-aerosol" sulphate formation by oxidation of dissolved $SO_2$. Even though the solubility of $O_3$ is quite low, the effective uptake is determined by the reaction rate.

Reply: Regarding the sulphate oxidation in ECHAM-HAMMOZ. We don't take into account $SO_2$ oxidation in aerosol phase (only in cloud droplets) as a chemical process. However, the oxidation is considered so, that part of $SO_2$ is emitted as primary sulphate. This approximation represents the aerosol phase oxidation of $SO_2$ in ECHAM-HAMMOZ. Similarly, in EMEP, $SO_2$ oxidation to sulphate on aerosols is assumed to take in gas and cloud rather than via aerosol water. As with ECHAM, a certain percentage (5%) of S emissions are assumed to be as sulphate.

7) What is the tropospheric $CH_4$ lifetime in your simulations? Due to $HO_2$ uptake, the oxidation capacity might be reduced, but due to HONO formation and subsequent photolysis otherwise affected. Is this similar in both models? Is there a substantial impact on the oxidation capacity

Table 3: Sensitivity Study: As in Table 4 (main manuscript), but with $\gamma(N_2O_5)$ set to either 0.01 or 0.1. calculations with EMEP model, year 2012.

| Region | Run | Unit | $O_3$ | $NO_x$ | $NO_y$ | $HNO_3$ | PAN | $N_2O_5$ | $NO_3$ |
|--------|-----|------|-------|--------|--------|---------|-----|----------|--------|
| Test 1: $\gamma(N_2O_5) = 0.01$ | | | | | | | | | |
| NA | Base | Conc*: | 41.18 | 0.84 | 1.83 | 0.20 | 0.57 | 7.35 | 5.62 |
| NA | noN2O5pm | %: | 3 | 6 | 3 | -7 | 5 | 80 | 29 |
| NA | noHNO3pm | %: | 0 | 0 | -2 | 18 | 0 | 0 | 1 |
| NA | noNO2pm | %: | 0 | 2 | 1 | -1 | 0 | 5 | 2 |
| EUR | Base | Conc*: | 41.89 | 1.06 | 2.46 | 0.24 | 0.56 | 12.75 | 7.88 |
| EUR | noN2O5pm | %: | 4 | 11 | 2 | -12 | 6 | 131 | 41 |
| EUR | noHNO3pm | %: | 1 | 0 | -6 | 60 | 0 | 0 | 3 |
| EUR | noNO2pm | %: | 0 | 5 | 1 | -1 | -1 | 7 | 4 |
| EA | Base | Conc*: | 45.00 | 2.28 | 4.68 | 0.53 | 0.92 | 16.98 | 7.05 |
| EA | noN2O5pm | %: | 5 | 12 | 3 | -16 | 9 | 180 | 61 |
| EA | noHNO3pm | %: | 0 | 0 | -2 | 13 | 0 | 0 | 1 |
| EA | noNO2pm | %: | -1 | 30 | 9 | -12 | -7 | 16 | 5 |
| SA | Base | Conc*: | 48.05 | 1.15 | 2.91 | 0.41 | 0.33 | 11.95 | 13.39 |
| SA | noN2O5pm | %: | 5 | 9 | 0 | -3 | 12 | 107 | 47 |
| SA | noHNO3pm | %: | 1 | 0 | -8 | 62 | 0 | 1 | 4 |
| SA | noNO2pm | %: | 1 | 4 | 1 | 0 | 1 | 10 | 6 |
| Test 2: $\gamma(N_2O_5) = 0.1$ | | | | | | | | | |
| NA | Base | Conc*: | 39.30 | 0.79 | 1.78 | 0.21 | 0.53 | 2.91 | 3.24 |
| NA | noN2O5pm | %: | 8 | 13 | 6 | -14 | 12 | 354 | 123 |
| NA | noHNO3pm | %: | 0 | 0 | -2 | 17 | 0 | 0 | 1 |
| NA | noNO2pm | %: | 0 | 2 | 1 | 0 | 0 | 3 | 2 |
| EUR | Base | Conc*: | 39.69 | 0.97 | 2.40 | 0.26 | 0.52 | 4.20 | 4.55 |
| EUR | noN2O5pm | %: | 10 | 21 | 5 | -20 | 14 | 600 | 145 |
| EUR | noHNO3pm | %: | 1 | 0 | -6 | 56 | 0 | 1 | 4 |
| EUR | noNO2pm | %: | 0 | 5 | 1 | 0 | -1 | 5 | 4 |
| EA | Base | Conc*: | 42.83 | 2.15 | 4.58 | 0.56 | 0.86 | 8.25 | 3.95 |
| EA | noN2O5pm | %: | 11 | 19 | 6 | -22 | 16 | 477 | 188 |
| EA | noHNO3pm | %: | 0 | 0 | -2 | 13 | 0 | 0 | 1 |
| EA | noNO2pm | %: | -1 | 30 | 9 | -10 | -8 | 10 | 3 |
| SA | Base | Conc*: | 45.80 | 1.06 | 2.87 | 0.43 | 0.31 | 7.18 | 8.57 |
| SA | noN2O5pm | %: | 10 | 18 | 2 | -7 | 22 | 245 | 129 |
| SA | noHNO3pm | %: | 1 | 0 | -8 | 60 | 0 | 1 | 4 |
| SA | noNO2pm | %: | 1 | 4 | 1 | 0 | 1 | 9 | 5 |

Notes: as in manuscript Table 4.

Table 4: Sensitivity Study: As in Table 4 (main manuscript), but with $\gamma(NO_2)$ set to either 0.001 or zero. calculations with EMEP model, year 2012.

| Region | Run | Unit | $O_3$ | $NO_x$ | $NO_y$ | $HNO_3$ | PAN | $N_2O_5$ | $NO_3$ |
|--------|-----|------|-------|--------|--------|---------|-----|----------|--------|

Test 3: $\gamma(NO_2) = 0.001$

| Region | Run | Unit | O3 | NOx | OXN | HNO3 | PAN | N2O5 | NO3 |
|--------|-----|------|-----|------|------|------|-----|------|------|
| TEST3 | | | | | | | | | |
| NA | Base | Conc*: | 39.16 | 0.72 | 1.73 | 0.21 | 0.54 | 3.88 | 3.85 |
| NA | noN2O5pm | %: | 4 | 6 | 3 | -6 | 6 | 113 | 45 |
| NA | noHNO3pm | %: | 0 | 0 | -2 | 17 | 0 | 0 | 1 |
| NA | noNO2pm | %: | 3 | 16 | 5 | -4 | 2 | 36 | 21 |
| | | | | | | | | | |
| EUR | Base | Conc*: | 39.66 | 0.83 | 2.31 | 0.26 | 0.54 | 5.33 | 4.88 |
| EUR | noN2O5pm | %: | 5 | 9 | 2 | -9 | 9 | 191 | 59 |
| EUR | noHNO3pm | %: | 1 | 0 | -7 | 57 | 0 | 0 | 2 |
| EUR | noNO2pm | %: | 3 | 28 | 6 | -4 | -1 | 53 | 38 |
| | | | | | | | | | |
| EA | Base | Conc*: | 42.95 | 1.39 | 4.12 | 0.64 | 0.94 | 7.05 | 4.45 |
| EA | noN2O5pm | %: | 5 | 5 | 2 | -5 | 9 | 127 | 62 |
| EA | noHNO3pm | %: | 0 | 0 | -2 | 11 | 0 | 0 | 0 |
| EA | noNO2pm | %: | 1 | 109 | 22 | -25 | -13 | 102 | 28 |
| | | | | | | | | | |
| SA | Base | Conc*: | 44.70 | 0.89 | 2.73 | 0.42 | 0.30 | 5.31 | 7.98 |
| SA | noN2O5pm | %: | 3 | 5 | 1 | -2 | 7 | 58 | 31 |
| SA | noHNO3pm | %: | 1 | 0 | -8 | 60 | 0 | 0 | 4 |
| SA | noNO2pm | %: | 7 | 32 | 7 | -2 | 11 | 114 | 59 |

Test 4: $\gamma(NO_2) = 0.0$

| Region | Run | Unit | O3 | NOx | OXN | HNO3 | PAN | N2O5 | NO3 |
|--------|-----|------|-----|------|------|------|-----|------|------|
| NA | Base | Conc*: | 40.50 | 0.84 | 1.82 | 0.20 | 0.55 | 5.27 | 4.63 |
| NA | noN2O5pm | %: | 6 | 9 | 5 | -11 | 9 | 170 | 62 |
| NA | noHNO3pm | %: | 0 | 0 | -2 | 18 | 0 | 0 | 0 |
| | | | | | | | | | |
| EUR | Base | Conc*: | 41.03 | 1.06 | 2.46 | 0.25 | 0.54 | 8.17 | 6.73 |
| EUR | noN2O5pm | %: | 7 | 17 | 4 | -18 | 11 | 304 | 75 |
| EUR | noHNO3pm | %: | 1 | 0 | -6 | 58 | 0 | 0 | 3 |
| | | | | | | | | | |
| EA | Base | Conc*: | 43.59 | 2.90 | 5.04 | 0.48 | 0.82 | 14.23 | 5.72 |
| EA | noN2O5pm | %: | 8 | 21 | 7 | -26 | 11 | 336 | 120 |
| EA | noHNO3pm | %: | 0 | 0 | -2 | 15 | 0 | 0 | 0 |
| | | | | | | | | | |
| SA | Base | Conc*: | 47.67 | 1.17 | 2.93 | 0.41 | 0.33 | 11.37 | 12.67 |
| SA | noN2O5pm | %: | 7 | 12 | 1 | -5 | 16 | 160 | 70 |
| SA | noHNO3pm | %: | 1 | 0 | -8 | 62 | 0 | 1 | 4 |

Notes: as in manuscript Table 4.

|         | REF  | noN2O5 | noHNO3 | noHO2 | noNO2 | noNO3 | noO3 |
|---------|------|--------|--------|-------|-------|-------|------|
| $\tau_{CH_4}$ | 8.4  | 8.1    | 7.7    | 8.3   | 8.3   | 8.3   | 8.4  |
| [OH]    | 1.02 | 1.05   | 1.10   | 1.03  | 1.03  | 1.02  | 1.02 |

Table 5: $CH_4$ lifetime and air volume weighted global annual mean tropospheric OH concentration in $10^6$ molec cm$^{-3}$ from ECHAM-HAMMOZ.

which is not that obvious in the well buffered compound $O_3$?

Reply: Table 5 shows $CH_4$ lifetimes and mean OH concentrations for each run done with ECHAM-HAMMOZ. As can be seen the $CH_4$ lifetime is 8.4 years in the reference run and gets shortened by more than a month turning off $N_2O_5$, $HNO_3$ reactions. The impact of $HNO_3$ reaction has strong effect in ECHAM-HAMMOZ, as discussed in the manuscript. Activation of $HO_2$ reaction prolongs methane lifetime by 22 days. This means that $HO_2$ loss has a net effect reducing the oxidative capacity. $NO_2$ reactions have a comparable effect, but less strong effect (just 4 days). Changes in methane lifetime are mirrored in annual mean OH concentrations, which are lowest when methane lifetime is largest. The EMEP model does not diagnose $CH_4$ lifetimes, so we cannot readily provide equivalent information. (For information, calculations made some years ago suggest however a lifetime of about 9 years for the EMEP model; M. Gauss, Pers. Comm., 2017). Given that changes in $CH_4$ are somewhat beyond the near-ground focus on ozone and short-lived gases of our other results, and we cannot compare the models, we think it better not to bring this issue up in the manuscript.

**1.3   Technical comments**

1) Fig. 6 to 8: The grey shaded area is very difficult to see. Please think about a better visualisation of the observations.

Reply: Plots with lines instead of shading for the observations are hard to read, but we will change the colour of the shading to improve visibility.

2) page 7, last line: citation is missing/wrong (ibid)

Reply: Changed

3) page 13, line 7 (wrong line number): citation is missing/wrong (ibid) "transport model, GEOS-Chem (ibid),....."

Reply: Changed

**Extra references**

Dentener, F. J. and Crutzen, P. J.: Reaction of $N_2O_5$ on tropospheric aerosols: Impact on the global distributions of $NO_x$, $O_3$, and OH, Journal of Geophysical Research: Atmospheres (1984–2012), 98, 7149–7163, 1993.

Tørseth, K., Aas, W., Breivik, K., Fjæraa, A. M., Fiebig, M., Hjellbrekke, A. G., Lund Myhre, C., Solberg, S., and Yttri, K. E.: Introduction to the European Monitoring and Evaluation Programme (EMEP) and observed atmospheric composition change during 1972–2009, Atmos. Chem. Phys., 12, 5447–5481, doi:10.5194/acp-12-5447-2012, URL `http://www.atmos-chem-phys.net/12/5447/2012/`, 2012.

---

## Author Comment (AC2) · 30 Nov 2017

**1 Anonymous Referee #2**

**1.1 General Comments**

This paper presents two global model simulations (EMEP, ECHAM-HAMMOZ) that assess the influence of six heterogeneous reactions on global atmospheric mixing ratios of reactive nitrogen species and ozone. The six reactions (given in Table 1) have been investigated in previous model studies, notably Jacob, Atmos. Environ. 2000, on which this submission appears to be largely based. The influence of heterogeneous chemistry is known to be important in global chemical transport models, but it is also generally difficult to parameterize for a number of reasons. These include the difficulty of accurately simulating aerosol surface areas available for heterogeneous reactions and the large uncertainty in some uptake coefficients, especially $N_2O_5$, which has the largest effects in the analysis from this paper. For these reasons, further investigations of the details of the heterogeneous reactions in global models are generally valuable contributions to the literature.

Reply: We thank the referee for the positive comments. As noted further below, we believe our article presents material which has not been presented before for CTMs, but we agree about the uncertainties and a constant need to re-evaluate the importance of these reactions.

Before giving the main replies below, we should point out that for the revised manuscript we re-ran all results with an updated version of the EMEP model. This was done following some bug-fixes in the rv4.15 version originally used, including in the deposition of $N_2O_5$ and radiation schemes. These changes have not affected basic model performance very much, but especially the deposition change affects $N_2O_5$ levels, and the impact of the noN2O5 scenarios. Indeed, the impacts in the EMEP system now resemble much more those of ECHAM-HAMMOZ. We have added a small Appendix to explain these changes, and modified the manuscript to reflect the updated results.

While the above is a good justification for the present work, it is somewhat difficult to see that this paper advances the subject much beyond what has been presented in previous papers. This deficiency in presentation could likely be addressed, but the authors would do well to review how their results compare to previous model simulations that have investigated this set of

reactions previously, as well as explicitly stating how their treatment differs and why their model arrives at different results or corroborates previous analysis. Such a comparison is and critical model evaluation is absent from the discussion section. The paper would be much stronger if it were included.

Reply: To strengthen our results, we included an additional Discussion section (Sect. 5.4) (also following your comment below about 18 lines 4-11). We find, $N_2O_5$ hydrolysis on ozone and nitrogen oxides is lower compared to previous studies although the other studies used rather high $\gamma$ values for $N_2O_5$ and older CTMs, and they neglected the other heterogeneous reactions. Details can be found in the reply below.

There is no discussion of the effects of clouds, which have large surface areas for heterogeneous reactions. Are all simulations showing effects of reactions on aerosols but not in clouds? This should be clarified, together with some estimate of the relative effects of both cloud droplets and aerosols if they are both operative in the models.

Reply: The simulations shown here are not including heterogeneous reactions on cloud surfaces, which can be important especially for $HO_2$ uptake depending of the presence of transition metal ions. The large surface area provided by clouds would lead to unrealistically high uptake rates, because the current parameterizations in EMEP and ECHAM-HAMMOZ do not include diffusion limitation (Davidovits, 2006). Often global model studies exclude the heterogeneous reactions of nitrogen species on clouds, for $N_2O_5$ Dentener and Crutzen (1993) included the reaction on cloud droplets, but just found minor changes in $NO_x$ and $O_3$. Jacob (2000) argues that for $O_3$, $HO_x$ and $NO_x$ life times are not significantly reduced in clouds and current knowledge is insufficient to include cloud chemistry in $O_3$ models. Therefore, the current model systems cannot treat heterogeneous cloud chemistry without further development and this is why we decided to not look at them in this study. We added some text on this in the new Sect. 5.4.

The paper identifies $N_2O_5$ uptake as the most important heterogeneous reaction of the six, but it does not include the production of $ClNO_2$ from $N_2O_5$ uptake. The authors state this deficiency clearly, but at that same time it is a missed opportunity since it would be one aspect where these model simulations could clearly take advantage of recent advances in field and laboratory work. No real explanation is given as to the technical details that prevent the inclusion of this reaction, but the omission should be better justified. Even a crude estimate of this reaction would be helpful to this analysis.

Reply: Yes, technical details was a little vague we admit! The problem is that neither model handles the treatment of Cl in the aerosol thermodynamics. Nor do they handle the chemistry of $ClNO_2$ and associated precursors and products. This makes it very difficult to come up with even a crude estimate of this reaction. We have made these limitations clearer in the manuscript, and strengthened this need in the text discussing future studies. We added some text on this in the new Sect. 5.4.

Although the paper does not appear to represent a significant advance (unless the authors provide some further comparisons and details), it does not appear to be incorrect in any obvious way. There are some issues with presentation, detailed below, but these issues do not appear to be serious. With some attention to the comments above and the more specific comments below, it should be suitable for publication in ACP.

Reply: Again, we thank the referee for the constructive remarks. As noted also in our reply to Referee #1, we believe that our paper is novel in using very up-to-date chemical transport models (CTMs) which we show can reproduce well daily variations at sites around the world, and with a demonstration of a fair ability to capture aerosol surface area compared to satellite data. In addition, we also illustrate in detail how seasonal patterns are affected by this reaction. We also believe we are the first to systematically compare the impacts of the different reactions in a harmonized way across two model systems. We have now added also a comparison to previous models studies and also new sensitivity tests on the impact of different assumptions for $\gamma$. Details can be found in the new Discussion Section 5.4.

**1.2 Specific Comments**

Page 2, line 1: Solomon et al., Nature 1986 is a better reference (suggestion only)

Reply: We added this reference.

Page 2, line 3: Ravishankara 1997 is a better reference here (again, suggestion only). Ravishankara, A. R. (1997), Heterogeneous and multiphase chemistry in the troposphere, Science, 276, 1058-1065.

Reply: We added this reference.

Page 2, line 22: Not clear what is meant by technical limitations here that precludes the inclusion of $ClNO_2$.

Reply: Yes, We addressed this issue in our comment above.

Page 4, first paragraph: The term $S_a$ is used to refer to aerosol surface area, but later in the paragraph $S$ is used as total surface area. Is there a distinction between $S$ and $S_a$, or is this just typographic.

Reply: We corrected the paper to use $S_a$ throughout.

Page 4, line 17: correct grammar in make use

Reply: Corrected.

Page 5, lines 8-13: Does the lack of nitrate aerosol formation artificially reduce aerosol surface area available for heterogeneous chemistry?

Reply: There are several major compounds contributing to $S_a$, in particular sulphate, organic aerosol, black-carbon, sea-salt and dust, so $S_a$ should not be too sensitive to nitrate. And Figures 2 and 5 suggest that although nitrate aerosol lacks in ECHAM-HAMMOZ, the surface area density is comparable to van Donkelaars estimate based on satellite data and to EMEP surface area. There are even regions where ECHAM-HAMMOZ calculates a higher surface area than EMEP.

Page 6, line 15: The value of 5e-1 (0.5) must be misquoted as the $N_2O_5$ reaction probability is not this large.

Reply: Corrected; this should have been (0.5- 6) x $10^{-6}$. We have made it explicit

Page 7, lines 14-16: The authors make a good point regarding the reliability of the parameterization, especially in light of the absence of atmospheric determinations of gamma values as high as those shown in figure 1. Have the authors made any assessment of the effect of reducing the RH dependence of the parameterization for sulfate?

Reply: Although we suspect the RH dependence is unrealistic, we have no real basis to devise an alternative formulation. Such information would need new laboratory or theoretical suggestions.

Page 7, section 3.2: The literature cited for the gamma value of $NO_3$ is dated. More recent work by Gross et al. shows substantial reactivity on organics, for example. There is somewhat less active research in this area than for uptake coefficients for $N_2O_5$, so there is no developed parameterization. It is not realistic for the authors to undertake such a review in the context of this paper, but some reference to the more recent studies together with a statement that $NO_3$ uptake may be larger if organic aerosol is considered is needed in this section.

Reply:    The missing literature has now been included in the manuscripts small literature overview, Sect. 3.2, see following paragraph.

*The nitrate radical $NO_3$ undergoes hydrolysis in wet aerosols, but was also observed to react with organic compounds on the aerosol surface. Hydrolysis of the nitrate radical $NO_3$ happens on various aerosol types depending on the water content. $NO_3$ heterogeneous reaction produces $HNO_3$ and OH in the aqueous particle phase and can be counted as a $NO_x$ sink (Rudich et al. 1998). Several laboratory studies shown $\gamma$ ranging between $10^{-4}$ and $10^{-3}$ (Rudich et al. 1996, Moise et al. 2002). Jacob (2000) recommended to use $\gamma = 10^{-3}$ for atmospheric chemistry model simulations.*

*Reactions with different organic compounds were explored in laboratory experiments. Gross and Bertram (2008) measured the reaction probabilities between 0.059 and 0.79 of $NO_3$ with different polycyclic aromatic hydrocarbons leading to $NO_2$ and $HNO_3$ formation. Two following studies also found high reaction probabilities of $NO_3$ with alkenoic acid ($>$0.07) (Gross et al. 2009) and alkene monolayers (0.034) (Gross and Bertram 2009). Organic coatings could enhance $NO_3$ reactive uptake, nevertheless knowledge of explicit organic compounds in the organic fraction of aerosol is unknown in both model systems, therefore the recommended of $\gamma = 10^{-3}$ for $NO_3$ hydrolysis value was adopted for EMEP and ECHAM-HAMMOZ.*

Page 7, section 3.3: Again, the authors are justified in the use of the simple uptake coefficient for $NO_2$ based on what is currently available in the literature, but the system is at least as complex as that for $N_2O_5$. Some model studies have assumed effectively very large uptake coefficients for $NO_2$ or at the very least rapid conversion of $NO_2$ to HONO. This body of literature should be represented here via referencing. One example of a recent modeling study: Elshorbany, Y. F., P. J. Crutzen, B. Steil, A. Pozzer, H. Tost, and J. Lelieveld (2014), Global and regional impacts of HONO on the chemical composition of clouds and aerosols, Atmos. Chem. Phys., 14(3), 1167-1184, 10.5194/acp-14-1167-2014.

Reply: We did not see a good way of including the results of the Elshorbany paper in our study. Elshorbany et al. 2014 looked at the impacts of a non-mechanistic and instantaneous gas-phase conversion of $NO_2$ to HONO (2 % yield) on the composition of aerosols and clouds, so they were not dealing with heterogeneous reactions as in our study. Such changes have been simulated by an enhanced gas-phase $H_2SO_4$ production caused by OH from HONO photolysis.

Page 13, section 5.1 and Figure 2: What altitude range is shown in Figure 2? Is this for some distance above the surface, boundary layer only, column average, etc?

Reply: All figures and tables refer to the lowest model layers, therefore the description ground level was added to the figures. In the model descriptions lowest layer thicknesses are mentioned. ECHAM-HAMMOZ lowest level has a thickness of 50 m, EMEP has 90 m, but EMEPs ground level refers to the value at 3 m height.

Second, is the displayed quantity a dry aerosol surface area or does it include water? If the latter (presumably), to what extent are the regional variations due to RH and to what extent to dry aerosol mass?

[Figure]

Figure 1: Fraction of dry aerosol over total aerosol mass containing water for the lowest model layer at ∼50 m over ground. If aerosol is wet in HAMMOZ the surface area is calculated based on the wet radius caused by the water content.

Reply: Indeed, the displayed aerosol surface area includes water, consistent with the satellite observation. Figure 1 shows the fraction of dry aerosol over total aerosol, the blue color shows areas where the dry aerosol mass equals the total mass, while brown colors indicate that the majority of the aerosol mass is made of water. Especially in Europe and East Asia 70-80 % of aerosol mass are water, so the large surface area is also caused by water. Nevertheless, in East Asia also dry aerosol mass concentrations are high. Most heterogeneous reactions implemented into EMEP and ECHAM-HAMMOZ are hydrolysis reactions and need aerosol water, therefore our comparison including aerosol water to water containing observed aerosol fits the purpose well.

Table 4: Caption states that reference runs values are given in total mixing ratios. This term is not clear. Do the authors mean average? Once again, over what altitude range do these values apply? Was this information given elsewhere? If so, it should be repeated here as it is not clear when reading the table or figure 2. The caption also appears to be logically in error: Since the sensitivity runs were subtracted from the reference run, positive values mean higher mixing ratios in the reference run than in the sensitivity runs and vice versa. By this logic, a higher value in the reference run would lead to a negative displayed value, consistent with what is shown in the table (e.g., removing $O_3$ uptake should increase $O_3$ in the sensitivity run, leading to sensitivity > reference, or reference  sensitivity < 0, as shown for no $O_3$). It is also notable in this table that the change in $N_2O_5$ are larger than the total or average. How can this be?

Reply: We should have used the term regional average near-surface mixing ratios, and will do this in the revised manuscript. We do not think the original results were logically in error (we used REF-TEST there), but to further increase clarity we have now used percentage changes in Tables 4 and 5, and provided values of TEST - REF so that positive values show the increase in mixing ratios caused by turning off the individual reactions. With this we hope to make it easier for the reader to get a quick impression of the results. From the percentages it is visible that sometimes the changes are higher than the annual field average surface mixing ratio, like in the ECHAM-HAMMOZ case of Europe $N_2O_5$ mixing ratio changing by 177 %. Some examples of this can also be seen in Fig. 5, where e.g. the change in European or East Asian $N_2O_5$ are bigger than the base-case values.

The tables are somewhat difficult to interpret since they are in absolute units. Relative changes (e.g., -10at least be given in addition to the absolute changes, and could be substituted for them easily since the absolute value is given in each case for the reference run.

Reply: We changed the Tables and text accordingly - the revised style of Table is given as Tables 1-4 in the reply to Ref #1. The current tables have been moved to the supplement, because we think they still include valuable information.

Page 17, line 25: The meaning if the sentence is not clear. $NO_3$ rapidly photolyses, and resulting $NO_2$ likewise, so has a high ozone-formation potential.

Reply: In order to clarify the meaning of this sentence following lines were added to the manuscript:

*$NO_3$ rapidly photolyses and produces $NO_2$ and atomic oxygen $O_3(^3P)$. $NO_2$ subsequently photolyses and results in NO and a second $O_3(^3P)$. From these two reactions two ozone molecules can be formed, therefore $NO_3$ has a high ozone-formation potential.*

Figure 3,4: Again, please specify the altitude range in the captions and text.

Reply: Added.

Page 18, lines 4-11: How do the reductions in $O_3$ and $NO_x$ compare with those determined from other model studies, e.g., Dentener and Crutzen (1993), Tie et al. (2001, 2003), Alexander et al. (2009), Macintyre and Evans (2010) etc. Critical comparisons of these results to these and other literature studies are missing, but extremely important to place the current work in context and understand what advances have been made in this model analysis.

Reply: We have added text on such comparisons into a new section (5.4), focussing on $NO_x$ and $O_3$ % reductions, which will include the following points among other items of discussion:

The results from this study, suggesting that the $N_2O_5$ reactions is generally the most important among the heterogeneous reactions implemented tested into ECHAM-HAMMOZ and EMEP models are loosely consistent with results from previous studies (e.g. Dentener and Crutzen 1993, Tie et al. 2001, 2003, Alexander et al. 2009, Macintyre and Evans 2010), although the magnitude of the changes seems to be somewhat less with our models. Direct comparison is difficult since all studies report different metrics, domains, seasons, and years, usually underlining the higher importance of heterogeneous reactions in the aerosol loaded northern hemisphere. As shown before, the impact of $N_2O_5$ hydrolysis is higher during winter and Dentener and Crutzen (1993) report about a 75 % $NO_x$ and 20 % $O_3$ reduction in their winter (November-April) period. Similar $NO_x$ reductions are found by Tie et al. (2001) and Tie et al. (2003) with 73 % and up to 90 % in December (although Tie et al. (2003) reports maximum reductions, whilst Tie et al. (2001) reports averages). In contrast, our models produce much smaller changes, for example ECHAM-HAMMOZ simulates a $NO_x$ reduction due to $N_2O_5$ hydrolysis of 16 % in winter (December-February). Also, $O_3$ reductions with our models are somewhat lower compared to these other models: e.g. 8 % in ECHAM-HAMMOZ compare to 11 % to 20 % in the other three model studies.

There are many possible reasons for these differences. Firstly, there have been many changes in models, emissions, and indeed the atmosphere since these early studies. For example, Dentener and Crutzen (1993) provided one of the first quantifications of the importance of $N_2O_5$ reactions, but the model used had a horizontal resolution of $10 \times 10°$, giving grid cells with 100 times the area of the $1 \times 1°$ grid used in EMEP or almost 30 times that of ECHAM-HAMMOZs $1.85 \times 1.85°$ grid. This alone will lead to different regimes of ozone productivity. For example, the collectio of earlier models included in the multi-model comparison of Wild et al. (2012) showed a very wide-spead of ozone results for the Mace Head site. Our comparisons (Fig. 7 in manuscript) suggest significant progress in model performance. Emissions have also changed enormously over this period, especially in Asia (Granier et al. 2011); again with implications for the atmospheric oxidation capacity.

The $\gamma_{N_2O_5}$ values used by Dentener and Crutzen 1993 and Tie et al. (=0.1) are significantly larger than the mean value of 0.02 calculated globally by the parametrizations used here, or than atmospheric observations (Brown et al. 2009)). Macintyre and Evans (2010) tested the model sensitivity to uniform $\gamma_{N_2O_5}$ values and report the highest sensitivity between 0.001 and 0.02. This is exactly the range of values given by the $\gamma_{N_2O_5}$ parametrization used here. As discussed below, and illustrated in Table 3 of the reply to Ref #1, the impact of the hydrolysis reaction on ozone is indeed stronger with higher $\gamma$, but our main results are relatively insensitive to these necessarily very uncertain choices. The ECHAM-HAMMOZ and EMEP models also have a set of other heterogeneous reactions competing with $N_2O_5$ hydrolysis, which again lowers the possible impact of this hydrolysis reaction.

As noted above, chemical transport models have developed in many ways over the last 20-30 years, and in this paper we show that both ECHAM-HAMMOZ and EMEP can reproduce even daily ozone variations at sites across the globe. We have also demonstrated that both models do a fair job of reproducing surface area density, so we believe our new estimates provide a valuable up-to-date revision of these earlier calculations.

As with Tables 4 and 5, these figures would be more easily interpreted in relative units (% change in $O_3$) rather than absolute units (ppbv) as shown.

Reply:   Tables 4 and 5 were updated showing annual mean surface mixing ratios in ppb (or ppt) for the reference runs and percentage changes between the reference run and the sensitivity runs - the revised style of Table is given as Tables 1-4 in the reply to Ref #1. However, we have retained the absolute numbers for maps since otherwise division by small numbers can produce rather confusing effects. With $NO_x$ especially some of the absolute values can be very low, but percentage changes very high.

Page 18, line 25: Suggest a change in the phrase therefore the sun is less favorable to something more like therefore photochemistry is inactive.

Reply: Changed.

Page 18, line 28: The statement is that less ozone production reduction occurs (awkward phrasing).

Reply: For clarification the text was adjusted with the following lines:

*During winter, nights are longer leading to inactive photochemistry. Therefore, heterogeneous chemistry is efficient. Nevertheless, a rather inactive photochemistry also leads to less ozone production. Comparing to spring, the impact seen here is lower because of already low ozone formation rate.*

Is this a statement about absolute or relative ozone production? The latter would be more relevant, since it is already understood that ozone photochemistry is weaker in winter.

Reply: This statement is about absolute and relative ozone production,although the differences are actually quite modest, eg. 7.7% O3 reductionfor HAMMOZ in winter (DJF) compared to 9.2% reduction in spring (MAM). We added this information to the text.

Page 22, line 9: The comma should be after ground stations. Having it before ground stations changes the meaning of the sentence in a way that the authors probably do not intend.

Reply: Changed.

**Extra references**

Alexander, B., Hastings, M., Allman, D., Dachs, J., Thornton, J., and Kunasek, S.: Quantifying atmospheric nitrate formation pathways based on a global model of the oxygen isotopic composition ($\Delta$ 17 O) of atmospheric nitrate, Atmospheric Chemistry and Physics, 9, 5043–5056, 2009.

Brown, S. S., Dube, W. P., Fuchs, H., Ryerson, T. B., Wollny, A. G., Brock, C. A., Bahreini, R., Middlebrook, A. M., Neuman, J. A., Atlas, E., Roberts, J. M., Osthoff, H. D., Trainer, M., Fehsenfeld, F. C., and Ravishankara, A. R.: Reactive uptake coefficients for $N_2O_5$ determined from aircraft measurements during the Second Texas Air Quality Study: Comparison to current model parameterizations, J. Geophys. Res. Atmos., 114, D00F10, doi: 10.1029/2008JD011679, 2009.

Dentener, F. J. and Crutzen, P. J.: Reaction of $N_2O_5$ on tropospheric aerosols: Impact on the global distributions of $NO_x$, $O_3$, and OH, Journal of Geophysical Research: Atmospheres (1984–2012), 98, 7149–7163, 1993.

Granier, C., Bessagnet, B., Bond, T., D'Angiola, A., van der Gon, H. D., Frost, G. J., Heil, A., Kaiser, J. W., Kinne, S., Klimont, Z., Kloster, S., Lamarque, J.-F., Liousse, C., Masui, T., Meleux, F., Mieville, A., Ohara, T., Raut, J.-C., Riahi, K., Schultz, M. G., Smith, S. J., Thompson, A., van Aardenne, J., van der Werf, G. R., and van Vuuren, D. P.: Evolution of anthropogenic and biomass burning emissions of air pollutants at global and regional scales during the 1980-2010 period, Climatic Change, 109, 163–190, doi:10.1007/s10584-011-0154-1, 2011.

Gross, S. and Bertram, A. K.: Reactive uptake of $NO_3$, $N_2O_5$, $NO_2$, $HNO_3$, and $O_3$ on three types of polycyclic aromatic hydrocarbon surfaces, The Journal of Physical Chemistry A, 112, 3104–3113, 2008.

Gross, S. and Bertram, A. K.: Products and kinetics of the reactions of an alkane monolayer and a terminal alkene monolayer with $NO_3$ radicals, Journal of Geophysical Research: Atmospheres, 114, 2009.

Gross, S., Iannone, R., Xiao, S., and Bertram, A. K.: Reactive uptake studies of $NO_3$ and $N_2O_5$ on alkenoic acid, alkanoate, and polyalcohol substrates to probe nighttime aerosol chemistry, Physical Chemistry Chemical Physics, 11, 7792–7803, 2009.

Jacob, D. J.: Heterogeneous chemistry and tropospheric ozone, Atmospheric Environment, 34, 2131–2159, 2000.

Macintyre, H. and Evans, M.: Sensitivity of a global model to the uptake of $N_2O_5$ by tropospheric aerosol, Atmospheric Chemistry and Physics, 10, 7409–7414, 2010.

Moise, T., Talukdar, R., Frost, G., Fox, R., and Rudich, Y.: Reactive uptake of $NO_3$ by liquid and frozen organics, Journal of Geophysical Research: Atmospheres (1984–2012), 107, AAC–6, 2002.

Rudich, Y., Talukdar, R. K., Ravishankara, A., and Fox, R.: Reactive uptake of $NO_3$ on pure water and ionic solutions, Journal of Geophysical Research: Atmospheres (1984–2012), 101, 21 023–21 031, 1996.

Rudich, Y., Talukdar, R., and Ravishankara, A.: Multiphase chemistry of $NO_3$ in the remote troposphere, Journal of Geophysical Research: Atmospheres (1984–2012), 103, 16 133–16 143, 1998.

Tie, X., Brasseur, G., Emmons, L., Horowitz, L., and Kinnison, D.: Effects of aerosols on tropospheric oxidants: A global model study, Journal of Geophysical Research: Atmospheres, 106, 22 931–22 964, 2001.

Tie, X., Emmons, L., Horowitz, L., Brasseur, G., Ridley, B., Atlas, E., Stround, C., Hess, P., Klonecki, A., Madronich, S., et al.: Effect of sulfate aerosol on tropospheric NOx and ozone budgets: Model simulations and TOPSE evidence, Journal of Geophysical Research: Atmospheres, 108, 2003.

Wild, O., Fiore, A. M., Shindell, D. T., Doherty, R. M., Collins, W. J., Dentener, F. J., Schultz, M. G., Gong, S., MacKenzie, I. A., Zeng, G., Hess, P., Duncan, B. N., Bergmann, D. J., Szopa, S., Jonson, J. E., Keating, T. J., and Zuber, A.: Modelling future changes in surface ozone: a parameterized approach, Atmospheric Chemistry and Physics, 12, 2037–2054, doi: 10.5194/acp-12-2037-2012, 2012.

---

## Author Response (AR1)

**Manuscript changes in Ozone Impacts of Gas-Aerosol Uptake in Global Chemistry Transport Models by Scarlet Stadtler et al.**

We have modified the manuscript as described in our replies to the two referees. The major changes are described below. These changes (and also minor changes in text for clarification) are highlighted in the pdf with a blue colour for new additions, and a red crossed-out text for deletions.

The main changes can be summarised:

1. A new discussion Section 5.4 was added to address the comments of both referees concerning comparison to other global model studies and current model limitations for reactive uptake parameterizations. There also the technical limitations mentioned in the introduction are explained. (As a result of this we have deleted some of earlier comments about model limitations, since these are now addressed in Sect. 5.4.)

2. Section 5.2 was adjusted to include Tables 4 and 5 in percentage changes as requested. Therefore, the text also includes percentage values. As we think our original Tables with absolute concentration changes are still valuable, we have moved these to a new Supplementary information document.

3. Section 5.2 was expanded to include EMEP tests using different $\gamma$ values for $N_2O_5$ and $NO_2$ as requested by Referee #1.

4. Equation (3) in Section 3.1 was moved from text to Table 2, since in hindsight this makes the formulation clearer.

5. Plots in Section 5.3 were adjusted to be consisted with Tables 4 and 5 in terms of sign of the differences between the runs.

6. An updated EMEP model version (rv4.16) was used to produce the results used for this revised manuscript. The main results and conclusions are still the same, but all plots were adjusted showing the new EMEP data. This also lead to minor changes in the text describing Table 4. (In the Reply to Referees we had suggested moving this text to a new Appendix, but in revising the manuscript it seemed more logical to simply include the update details in the EMEP model description, Sect. 2.1.)

7. We added a supplementary material including tables with absolute differences, consistent with Table 4 and 5 in the manuscript. Furthermore, tables for new EMEP tests in Section 5.2 and a Table like Table 4 for 2011 can be found there. Also a plot showing annual mean surface $\gamma_{N_2O_5}$ values was added.